



# Atmospheric oxidation of α,β-unsaturated ketones: kinetics and mechanism of the OH radical reaction

Niklas Illmann[1], Rodrigo Gastón Gibilisco[2], Iustinian Gabriel Bejan[3], Iulia Patroescu-Klotz[1], and Peter Wiesen[1]

5 [1]Institute for Atmospheric and Environmental Research, Bergische Universität Wuppertal, 42097 Wuppertal, Germany
[2]INQUINOA-UNT-CONICET Institute of Physical Chemistry, Faculty of Biochemistry, Chemistry and Pharmacy, National University of Tucumán, San Lorenzo 456, T4000CAN, San Miguel de Tucumán, Argentina
[3]Faculty of Chemistry and Integrated Centre of Environmental Science Studies in the North Eastern Region Alexandru Ioan Cuza University of Iasi, 11 Carol I, Iasi 700506, Romania

*Correspondence to*: Niklas Illmann (illmann@uni-wuppertal.de)

**Abstract.** The OH radical initiated oxidation of 3-methyl-3-penten-2-one and 4-methyl-3-penten-2-one was investigated in two atmospheric simulation chambers at $298 \pm 3$ K and $990 \pm 15$ mbar using long-path FTIR spectroscopy. The rate coefficients of the reactions of 3-methyl-3-penten-2-one and 4-methyl-3-penten-2-one with OH radicals were determined to

15 be $(6.5 \pm 1.2) \times 10^{-11}$ cm$^3$ molecule$^{-1}$ s$^{-1}$ and $(8.1 \pm 1.3) \times 10^{-11}$ cm$^3$ molecule$^{-1}$ s$^{-1}$, respectively. To enlarge the kinetics data pool the rate coefficients of the target species with Cl atoms were determined to be $(2.8 \pm 0.4) \times 10^{-10}$ cm$^3$ molecule$^{-1}$ s$^{-1}$ and $(3.1 \pm 0.4) \times 10^{-10}$ cm$^3$ molecule$^{-1}$ s$^{-1}$, respectively. The mechanistic investigation of the OH initiated oxidation focuses on the $RO_2$ + NO reaction. The quantified products were acetoin, acetaldehyde, biacetyl, $CO_2$ and peroxyacetyl nitrate (PAN) for the reaction of 3-methyl-3-penten-2-one with OH radicals and acetone, methyl glyoxal, 2-hydroxy-2-methylpropanal, $CO_2$ and

20 peroxyacetyl nitrate (PAN) for the reaction of 4-methyl-3-penten-2-one with OH, respectively. Based on the calculated product yields an upper limit of 0.15 was determined for the overall organic nitrates ($RONO_2$) yield derived from the OH reaction of 4-methyl-3-penten-2-one. By contrast, no $RONO_2$ formation was observed for the OH reaction of 3-methyl-3-penten-2-one. Additionally, a simple model is presented to correct product yields for secondary processes.



## 1 Introduction

The α,β-unsaturated ketones are a particular class of oxygenated volatile organic compounds (OVOCs) emitted either from biogenic and/or anthropogenic sources or generated in the oxidation of airborne VOCs in the atmosphere. The most prominent representative among this class is methyl vinyl ketone (MVK). MVK is, on the one hand, emitted from polymer, pharmaceuticals and fungicides manufacturing processes industry (Siegel and Eggersdorfer, 2000). On the other hand, it is formed in the troposphere, mainly through the gas-phase oxidation of isoprene, the most abundant NMHC emitted into the atmosphere with an estimated annual emission up to 750 Tg (Calvert et al., 2000; Guenther et al., 2006). Other α,β-unsaturated ketones, like 3-methyl-3-penten-2-one (3M3P2) are used in the fragrances and food industry (Chapuis and Jacoby, 2001; Bickers et al., 2003, Wang et al., 2020). Furthermore, 4-methyl-3-penten-2-one (4M3P2), commonly known as mesityl oxide, is a precursor of methyl isobutyl ketone used extensively as solvent in the fabrication of paints and coatings. Production rates as high as $10^5$ t per year were reported in the US (Sifniades, 2011). α,β-unsaturated ketones were also identified in laboratory studies on emissions from different fuels representative for biomass burning (Hatch et al., 2017).

During daytime the main loss process of α,β-unsaturated ketones, once released into or formed in the atmosphere, is probably the oxidation initiated by OH radicals with direct impact on the atmospheric ozone and secondary organic aerosol formation (Kanakidou et al., 2005; Calvert et al., 2011). However, our knowledge about the oxidation mechanisms of species presented in Table 1 is rather limited. Up to date, only the OH radical reaction of MVK has been intensively studied, proving formaldehyde and methyl glyoxal as main oxidation products (Tuazon and Atkinson, 1987; Praske et al., 2015). Some attention was given also to the reaction of both 3M3P2 and 4M3P2, investigated in the present study, with $O_3$ and to the 3M3P2 + Cl and 3M3P2 + $NO_3$ system (Sato et al., 2004; Canosa-Mas et al., 2005; Wang et al., 2015; Illmann et al., 2021; Li et al., 2021). Gaona-Colmán et al. (2017) investigated the OH radical reaction of 4M3P2. They identified qualitatively though only formaldehyde and acetone (GC-MS detection). Hence, the mechanism remains rather incomplete.

In this work, we present an in-depth investigation of the OH radical initiated oxidation of two di-substituted α,β-unsaturated ketones, namely 3M3P2 and 4M3P2. Beside the first determination of the rate coefficient for the reaction of OH radicals with 3M3P2, we report kinetic data for Cl atom reactions.

In order to correct the formation yields of products formed in target reactions within complex experimental systems it is quite common to use the Tuazon formalism (Tuazon et al., 1986). This is based on the assumption that reaction products are subsequently consumed in secondary processes like photolysis, wall loss, and oxidation by OH radicals etc. Thus, their formation yields in the target reactions are underestimated when determined from plotting the formed product against the consumed compound of interest and the yields increase when corrected. However, the formalism cannot be applied for products with secondary sources in the reaction system. In this case, the formation yields are overestimated without proper corrections. For this purpose, we present a simple model to correct molar formation yields in the target reactions, which accounts for both consumption and secondary formation processes.



The study investigates the contribution of the OH initiated oxidation of both 3M3P2 and 4M3P2 to the formation of
$NO_x$ reservoir species, like peroxyacetyl nitrate (PAN), in the atmosphere. Apart for being an important $NO_x$ reservoir species
for the atmosphere, PAN is a phytotoxic air pollutant (Vyskocil et al. 1998) with still unaccounted sources (Fischer et al.,
2014). PAN formation depends strongly on temperature and the levels of $NO_2$ and NO. The study stresses the importance of
determining either the ratio between PAN and $CO_2$/HCHO or the PAN yield together with the $NO_2$/NO ratio within the
experiment since these give more comprehensive information on $NO_x$ reservoir species production than the PAN yield alone.
Therefore experiments were conducted under varying $NO_2$/NO ratios.

**Table 1.** Structures of α,β-unsaturated ketones and related literature on the corresponding OH radical initiated oxidation mechanism.

| Structure | $R_1$ | $R_2$ | $R_3$ | Name | Reference |
|---|---|---|---|---|---|
| | -H | -H | -H | Methyl vinyl ketone (3-buten-2-one) | Tuazon and Atkinson, (1989); Praske et al., (2015); Fuchs et al., (2018) |
| | -H | $-CH_3$ | -H | 3-Penten-2-one | Illmann et al., in preparation |
| | $-CH_3$ | -H | -H | 3-Methyl-3-buten-2-one | |
| | $-CH_3$ | $-CH_3$ | -H | 3-Methyl-3-penten-2-one | This work |
| | -H | $-CH_3$ | $-CH_3$ | 4-Methyl-3-penten-2-one | Gaona-Colmán et al., (2017); This work |


## 2 Experimental

Experiments were carried out in a 1080 L and a 480 L reaction chamber in 990 ± 15 mbar of synthetic air at 298 ± 3 K. In the
following section is given an updated description of both chambers. A major improvement of both chambers is the addition of
heatable injection blocks (< 100 °C). A controlling unit allows to adjust the temperature for a better transfer of samples into
the reaction chamber, according to the thermal stability of the investigated substances. Graphic representations of the chambers
were published already by Barnes and co-workers (Barnes et al., 1993; Barnes et al., 1994).

### 2.1 1080 L chamber

The 1080 L chamber consists of two joint quartz-glass tubes with a total length of 6.2 m and an inner diameter of 0.47 m
connected via a middle flange. It is closed at both ends by metal flanges bearing several ports for the injection of reactants,



addition of bath gases and coupling with analytical devices. 32 superactinic fluorescent lamps (Philips TL05 40W: 300–460 nm, max. intensity at ca. 360 nm) and 32 low-pressure mercury vapour lamps (Philips TUV 40W: max. intensity at 254 nm) can be used to irradiate the reaction mixture. These lamps are wired in parallel and spaced evenly around the reaction vessel. The pumping system consists of a turbo-molecular pump backed by a double-stage rotary fore pump. The chamber is

cleaned between experiments by evacuating it to $10^{-4}$ mbar for at least 30 minutes. Cleanliness is proved by FTIR. Homogeneity of the reaction mixtures is achieved by three magnetically coupled Teflon mixing fans, which are placed on the end and middle flanges. A White-type mirror system is installed inside the chamber to monitor reaction mixtures via FTIR spectroscopy in the spectral range 4000–700 $cm^{-1}$ and a resolution of 1 $cm^{-1}$. The system whose base length is $(5.91 \pm 0.01)$ m was operated at 82 traverses which yields a total optical path length of $(484.7 \pm 0.8)$ m. Spectra were recorded using a Nicolet

iS50 instrument equipped with a liquid nitrogen cooled mercury-cadmium-telluride (MCT) detector.

The initial mixing ratios in the 1080 L experiments, in ppmV (1 ppmV = $2.46 \times 10^{13}$ molecules $cm^{-3}$ at 298 K), were: 0.7–1.3 for 3-methyl-3-penten-2-one (3M3P2), 0.9–1.8 for 4-methyl-3-penten-2-one (4M3P2), 1.1 for isoprene, 0.9–1.4 for propene, 0.9–1.4 for isobutene, 0.9–1.4 for 1,3-butadiene, 0.9–1.9 for methyl nitrite, 0.9–1.9 for $Cl_2$, 2–4 for NO and 0–2.5 for $NO_2$.


## 2.2 480 L chamber

The smaller chamber in the Wuppertal laboratory consists of a borosilicate glass cylinder with a total volume of 480 L (length of 3 m and 0.45 m inner diameter). The tube is closed at both ends by aluminum flanges containing various ports for the introduction of reactants and bath gases, sampling and instruments monitoring the physical parameters inside the chamber. To

ensure homogeneous mixing of the reactants a magnetically coupled fan is mounted on the front flange inside the chamber. 32 fluorescent lamps (Philips TLA 40 W, $300 \leq \lambda \leq 460$ nm, $I_{max}$ = 360 nm) are mounted in four boxes and spaced evenly around the chamber. The lamps housings are cooled with air and their inner surface is encased in reflective steel sheets. The lamps can be switched individually to allow a variation of the photolysis frequency and consequently the radical level during photolysis experiments. The pumping system consists of a rotary vane pump and a roots pump yielding an end vacuum of up

to $10^{-3}$ mbar. For a typical cleaning procedure between two experiments the chamber is completely evacuated and filled up to 200–300 mbar of synthetic air or nitrogen. This procedure is repeated until it is certain that no signals related to the previous experiment are detected. Reactants and products are basically monitored using in-situ FTIR spectroscopy. For this purpose, a White-type mirror system (base length: $2.80 \pm 0.01$ m) is installed inside the chamber and coupled to a Nicolet 6700 FTIR spectrometer (MCT detector). The system is operated at 18 traverses which yields a total optical path length of $50.4 \pm 0.2$ m.

FTIR spectra are recorded in the spectral range 4000–700 $cm^{-1}$ and a resolution of 1 $cm^{-1}$.

The initial mixing ratios in the 480 L experiments in ppmV (1 ppmV = $2.46 \times 10^{13}$ molecules $cm^{-3}$ at 298 K) were: 5.0–6.1 for 3-methyl-3-penten-2-one (3M3P2), 5.0–6.0 for 4-methyl-3-penten-2-one (4M3P2), 5.0–5.7 for methyl valerate,



4.2–6.3 for propene, 4.2–6.3 for isobutene, 4.2–6.3 for 1,3-butadiene, 10–16 for methyl nitrite, 13–16 for Cl$_2$, and 20–27 for NO.


## 2.3 Materials

The following chemicals were used without further handling, with purities as stated by the suppliers: isobutene (Sigma Aldrich, 99%), propene (Air Liquide, 99.95%), 1,3–butadiene (Messer, >99%), isoprene (Aldrich, 99%), methyl valerate (Alfa Aesar, 99%), 3-methyl-3-penten-2-one (Sigma Aldrich, tech. 90%), 4-methyl-3-penten-2-one (Sigma Aldrich, tech. 90%), acetoin

(Sigma Aldrich, 96%), biacetyl (Sigma Aldrich, 97%), 2-methyl-3-buten-2-ol (Sigma Aldrich, 98%), carbon monoxide (Air Liquide, 99.97%), nitrogen monoxide (Air Liquide, 99.5%), nitrogen dioxide (Messer Griesheim, >98%), Cl$_2$ (Air Liquide, 99.8%), synthetic air (Messer, 99.9999%), nitrogen (Messer, 99.9999%). Methyl nitrite was synthesized by dropping sulphuric acid in an ice cooled aqueous solution of sodium nitrite and methanol (Taylor et al., 1980). The product was collected and stored in a cooling trap at -78 °C. The purity of methyl nitrite was confirmed via FTIR spectroscopy.


## 2.4 Experimental Protocol

OH radicals were generated by methyl nitrite photolysis in synthetic air at 360 nm:

$$CH_3ONO + h\nu \rightarrow CH_3O + NO \tag{R1}$$
$$CH_3O + O_2 \rightarrow HCHO + HO_2 \tag{R2}$$
$$HO_2 + NO \rightarrow OH + NO_2 \tag{R3}$$

NO was added to the reaction system to supress ozone formation and hence the formation of NO$_3$ radicals. Cl atoms were generated by photolysis of molecular chlorine in either synthetic air or nitrogen at 360 nm:


$$Cl_2 + h\nu \rightarrow 2\ Cl \tag{R4}$$

In both chambers, the target compound and the products formed during the reaction (mechanistic investigation), and the target and reference compound (kinetic study experiments) were monitored using FTIR spectroscopy. Typically, 50–70

interferograms were co-added per spectrum - which results in averaging period of about 80–113 s - and 15–20 spectra were recorded per experiment. In each run the first five spectra were collected in the dark, over a period of 10–20 min, to check for a potential wall loss of the unsaturated ketones and the reference compounds. Afterwards, the reaction was started by switching on the lamps. In the product study experiments, the reaction was terminated after the record of a maximum of 10 spectra. In



selected experiments additionally 5 spectra were collected in the dark, after termination, over a time interval of 10–20 min to
investigate the significance of the wall loss for the products of the oxidation reaction in the experimental system.

Usually, the housing which infolds the transfer optics between FTIR spectrometer and chamber is flushed with
purified dry air. Therefore, quantification of $CO_2$ is, due to a slight variability in the dry air supply, unreliable under normal
laboratory conditions. To be able to quantify $CO_2$, in selected product study experiments performed in the 1080 L chamber,
the transfer optics housing was flushed with ultrapure $N_2$ evaporated from a liquid nitrogen tank.


## 2.5 Relative rate method

The rate coefficients for the reaction of OH radicals and Cl atoms with the α,β-unsaturated ketones were determined by relating
the consumption of the ketone to the consumption of at least two reference compounds:

Ketone + OH → Products                                                                            (R5)
Reference + OH → Products                                                                            (R6)

Both, ketones and references, could potentially be subject of an irreversible first-order wall loss:

Ketone + wall → loss                                                                              (R7)
Reference + wall → loss                                                                            (R8)

Considering these processes the following equation can be derived:

$$\ln\left(\frac{[\text{ketone}]_0}{[\text{ketone}]_t}\right) - k_{\text{loss,ketone}} \times t = \frac{k_{ketone}}{k_{ref.}} \times \left(\ln\left(\frac{[\text{ref.}]_0}{[\text{ref.}]_t}\right) - k_{\text{loss,ref.}} \times t\right)$$                              (1)

where $[X]_t$ is the concentration of the species X at time t and t = 0 corresponds to the time where the lamps were switched on.
According to Eq. (1) a plot of $\left\{\ln\left(\frac{[\text{ketone}]_0}{[\text{ketone}]_t}\right) - k_{\text{loss,ref.}} \times t\right\}$ against $\left\{\ln\left(\frac{[\text{ref.}]_0}{[\text{ref.}]_t}\right) - k_{\text{loss,ref.}} \times t\right\}$ should yield a straight line
with zero intercept where the slope represents the rate coefficient ratio $k_{\text{ketone}}/k_{\text{ref.}}$.


## 2.6 Product identification and quantification

The quantification of identified products was done by subtraction of calibrated reference IR spectra. The corresponding
concentrations of the reference spectra were calculated based on cross sections either taken from literature references, from
the Wuppertal laboratory's database or determined within this work. An average value of the cross sections given by Profeta



et al. (2011) and Talukdar et al. (2011) has been used for methyl glyoxal. Peroxyacetyl nitrate (PAN) has been quantified using the absorption cross section reported by Allen et al. (2005). Cross sections for acetone and acetaldehyde were taken from the Wuppertal database. The absorption cross sections of 3-methyl-3-penten-2-one, 4-methyl-3-penten-2-one, 3-hydroxy-2-butanone (acetoin) and 2,3-butanedione (biacetyl) were determined within this work by either injecting different volumes of the pure compound into the chamber, evaporating weighted solid samples (in the case of acetoin) into a flow of bath gas or by

injecting aliquot volumes of a solution containing the target species according to a method previously described by Etzkorn et al. (1999). Reference spectra of 2-hydroxy-2-methylpropanal (HMPr) were generated in situ by the ozonolysis of 2-methyl-3-buten-2-ol in the presence of CO to scavenge any OH radical formed in the reaction system. $CO_2$ was quantified by integration of the absorption features in the range 2400–2349 $cm^{-1}$ and a polynomial calibration function derived from the injection of various volumes of $CO_2$ using a calibrated gas-tight syringe.


### 2.7 Modelling

In order to correct the experimentally determined product yields for both secondary formation and consumption a simple model was established based on the Euler-Cauchy method, which can be written down using a commercially available calculation program. In doing so, the rates d[X]/dt for each species X are calculated for constant time intervals Δt based on the rate equation

and the concentration of the species X in the previous time interval. This allows the calculation of concentration changes for each time interval, which, when added to the concentration of the previous time interval, yield the concentration of X at time t. The model assumes simplified reaction mechanisms as exemplarily shown below.

| | |
|---|---|
| A + OH → $y_b$ B + $y_c$ C + $y_d$ D | (R9) |
| 200   A + wall → loss | (R10) |
| B + OH → C | (R11) |
| C + OH → E | (R12) |
| C + wall → loss | (R13) |
| D + OH → $y_e$ E + $y_f$ F | (R14) |


The simplest version of this approach assumes a steady-state (sst) concentration of OH radicals that can be determined from the individual loss of the target species A during the irradiation period of each experiment. This hypothesis is valid as long as pseudo-first order conditions are proven by a linear correlation between {ln([A]$_0$/[A]$_t$) − $k_{wall}$ × t} and the time t. The modelled concentration time profiles are obtained by the input of [A]$_0$, [OH]$_{sst}$ and the rate coefficients of all listed reactions (R9–R14).

The yield $y_x$ for each product of the target reaction is included as parameter to be varied until the simulated concentration-time profile of each species matches the experimental data. However, for traceability and potential future applications, we prefer to outline in detail the stepwise procedure which turned out as best practice after several tests.





1.  The rate equations for each species are written according to the simplified exemplary mechanism shown above (R9–R14) where kinetic data and molar formation yields for the secondary reactions are taken from literature references. Values for the first-order rate coefficients of each species' wall loss are included if they are measured in the same experiment (thus measured before and after the irradiation of the reaction mixture). If they are not measured, values for $k_{wall}$ are first set to zero and included as additional variable. Molar formation yields for the target reaction are included as variable $y_x$.

2.  Time intervals $\Delta t$ and the total duration t are adjusted and the initial concentration $[A]_0$ of A is included for t = 0 according to the experimental data.

3.  If pseudo-first order conditions are proven by the experimental data, a constant OH concentration is included based on the consumption of A during the irradiation. If pseudo-first order conditions are not accomplished within the experiment the OH concentration has to be described as a function of t based on the consumption of A. However, including the OH concentration yields a simulated time profile for A which should well-reproduce the experimental data of A.

4.  Finally, the parameters $y_x$ for the target reaction are varied until the time profile of each species matches the experimental data, starting with $y_x$ corresponding to species X, which is the less affected by secondary reactions. Thus, one would start with $y_d$ and $y_b$ rather than $y_c$ in the given example. The uncorrected molar yields $y_x$, derived from plotting the measured mixing ratio of product X against the consumed target species A, are appropriate as starting point of the iterative process. If wall losses were not determined within the experiments one should try to match the time profile for the first data points where secondary processes are less significant. After that the wall loss parameters are varied to match the whole time profile. However, for this procedure it is mandatory to know the reasonable range of $k_{wall}$ for each species in the experimental set-up.

The time intervals used for the simulations were typically $\Delta t < 0.1$ s. The yields errors associated with the model were found to be in the range of 0.02. If it is not possible to reproduce the time profiles with the model this will indicate either an error in the spectra evaluation, the used simplified mechanism or the conducted experiments.

## 3 Results and discussion

All experiments were conducted at a total pressure of $990 \pm 15$ mbar and $298 \pm 3$ K. Irreversible first-order wall losses of 3M3P2 and 4M3P2 were found to be negligible in the 480 L chamber and in the range $(1–6) \times 10^{-5}$ s$^{-1}$ and $(1–8) \times 10^{-5}$ s$^{-1}$ in the 1080 L chamber, respectively. Dark reactions between the radical source and the target species were tested in separate experiments and were found to be negligible in both reaction chambers. Photolysis of the unsaturated ketones was likewise found to be negligible under the experimental conditions.





## 3.1 Kinetic study

Relative-rate plots according to Eq. (1) are presented in Figure 1 for all conducted kinetic experiments. Three reference compounds have been used to determine the rate coefficient of each investigated reaction system. The relative-rate plots for individual experiments display a high linearity, with correlation coefficients from linear regression analysis being $R^2 > 0.95$ and zero intercepts within a $2\sigma$ statistical error. The calculated relative ratios $k_{carbonyl}/k_{ref.}$ are summarised in Table 2. They were found to be independent of the initial concentration of the unsaturated ketone and the irradiation time. In the case of the Cl reactions, the relative ratios were independent of using either synthetic air or nitrogen as bath gas. Therefore, the obtained relative ratios likely result solely from each target reaction and any interfering process can be neglected in the present experimental set-up.

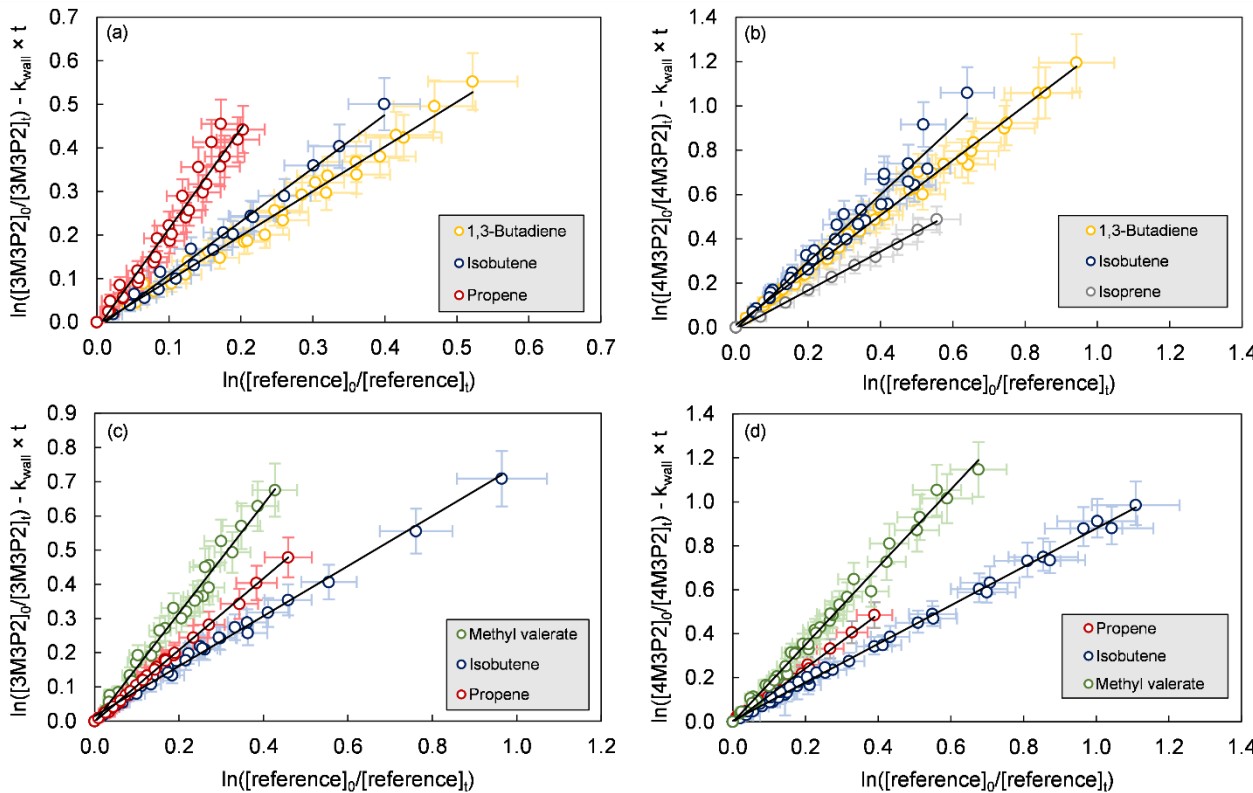

**Figure 1.** Relative-rate plots of all experiments according to equation (1) for the reaction of (a) 3-methyl-3-penten-2-one + OH, (b) 4-methyl-3-penten-2-one + OH, (c) 3-methyl-3-penten-2-one + Cl and (d) 4-methyl-3-penten-2-one + Cl. The error bars consist of a systematic uncertainty and an additional 10 % relative error to cover uncertainties derived from the experimental and evaluation procedure, respectively.

The rate coefficients were calculated based on the determined relative ratios using the following reference data: $k$(Isobutene + OH) = $(5.1 \pm 1.0) \times 10^{-11}$ cm$^3$ molecule$^{-1}$ s$^{-1}$ (IUPAC, current recommendation), $k$(Propene + OH) = $(2.9 \pm 0.8) \times 10^{-11}$ cm$^3$ molecule$^{-1}$ s$^{-1}$ (IUPAC, current recommendation), $k$(1,3-Butadiene + OH) = $(6.66 \pm 1.33) \times 10^{-11}$ cm$^3$ molecule$^{-1}$ s$^{-1}$





(Calvert et al., 2000), $k$(Isoprene + OH) = $(1.0 \pm 0.2) \times 10^{-10}$ cm$^3$ molecule$^{-1}$ s$^{-1}$ (IUPAC, current recommendation), $k$(Isobutene + Cl) = $(3.40 \pm 0.28) \times 10^{-10}$ cm$^3$ molecule$^{-1}$ s$^{-1}$ (Ezell et al., 2002), $k$(Propene + Cl) = $(2.64 \pm 0.21) \times 10^{-10}$ cm$^3$ molecule$^{-1}$ s$^{-1}$ (Ezell et al., 2002), $k$(Methyl valerate + Cl) = $(1.7 \pm 0.2) \times 10^{-10}$ cm$^3$ molecule$^{-1}$ s$^{-1}$ (Notario et al., 1998).


**Table 2.** Results of the kinetic study using different reference compounds.

| Reaction | Reference | Number of runs | Ketone [ppm]$_0$ | Ketone Max. consumption | $k_{\text{carbonyl}}/k_{\text{ref.}}$ | $k \times 10^{11}$ / cm$^3$ molecule$^{-1}$ s$^{-1}$ |
|---|---|---|---|---|---|---|
| 4M3P2 + OH | Isobutene | 4 | 5.0–6.0 | 47–65 % | 1.47 ± 0.31 | 7.5 ± 2.5 |
| | 1,3-Butadiene | 6 | 0.9–6.0 | 28–70 % | 1.29 ± 0.03 | 8.6 ± 1.9 |
| | Isoprene | 1 | 0.9 | 39 % | 0.88 ± 0.03 | 8.8 ± 1.3 |
| | | | | | **Average** | **8.1 ± 1.3** |
| 4M3P2 + Cl | Isobutene | 5 | 0.9–6.0 | 18–63 % | 0.90 ± 0.07 | 30 ± 3 |
| | Propene | 3 | 0.9–6.0 | 12–38 % | 1.18 ± 0.09 | 31 ± 4 |
| | Methyl valerate | 4 | 5.0–6.0 | 39–68 % | 1.79 ± 0.12 | 31 ± 4 |
| | | | | | **Average** | **31 ± 4** |
| 3M3P2 + OH | Propene | 3 | 0.7–6.1 | 34–37 % | 2.37 ± 0.40 | 6.9 ± 2.1 |
| | Isobutene | 2 | 0.7–6.1 | 21–39 % | 1.20 ± 0.02 | 6.1 ± 1.6 |
| | 1,3-Butadiene | 3 | 0.9–6.1 | 29–42 % | 1.01 ± 0.06 | 6.7 ± 1.4 |
| | | | | | **Average** | **6.5 ± 1.2** |
| 3M3P2 + Cl | Isobutene | 4 | 0.7–6.1 | 20–51 % | 0.80 ± 0.14 | 27 ± 5 |
| | Propene | 3 | 0.7–6.1 | 17–38 % | 1.07 ± 0.04 | 28 ± 3 |
| | Methyl valerate | 3 | 5.0–6.1 | 37–49 % | 1.60 ± 0.08 | 27 ± 3 |
| | | | | | **Average** | **28 ± 4** |

In Table 2 the calculated rate coefficients are given as an average for each reference together with the corresponding error as a combination of the relative ratio's statistical error and the error of the reference's rate coefficient. The determinations with

different reference compounds agree within < 4 % for the Cl reactions and < 17 % for the OH reactions, respectively. The final rate coefficients for the target reactions are given as the weighted average of all experimental determinations: $k$(3M3P2 + OH) = $(6.5 \pm 1.2) \times 10^{-11}$ cm$^3$ molecule$^{-1}$ s$^{-1}$, $k$(4M3P2 + OH) = $(8.1 \pm 1.3) \times 10^{-11}$ cm$^3$ molecule$^{-1}$ s$^{-1}$, $k$(3M3P2 + Cl) = $(2.8 \pm 0.4) \times 10^{-10}$ cm$^3$ molecule$^{-1}$ s$^{-1}$ and $k$(4M3P2 + Cl) = $(3.1 \pm 0.4) \times 10^{-10}$ cm$^3$ molecule$^{-1}$ s$^{-1}$. The quoted errors represent the 2σ statistical error of the weighted mean and an additional 10 % relative error to cover all uncertainties derived from the

experimental and evaluation procedure.

Rate coefficients for the reaction of 4M3P2 with OH radicals and Cl atoms have been previously determined by Gaona-Colmán et al. (2017), using the relative-rate technique and GC-FID as detection method. The present values are about 20 % smaller (OH reaction) and 15 % larger (Cl reaction) than reported by Gaona-Colmán et al. (2017). However, both values are found within the uncertainties of the former study, thus in good agreement. Wang et al. (2015) reported $k$(3M3P2 + Cl) =

$(3.00 \pm 0.34) \times 10^{-10}$ cm$^3$ molecule$^{-1}$ s$^{-1}$ at 293 ± 1 K based on a relative-rate study (GC-FID detection) employing large initial





mixing ratios for the target species (100 ppm) . Nevertheless, the value determined within this work is in excellent agreement with the one reported by Wang and co-workers.

### 3.1.1 Reactivity

It is generally accepted that OH radical and Cl atom reactions of OVOCs proceed via H atom abstraction or addition to the C=C double bond, in the case of unsaturated organic species. The AOPWIN software (US EPA, 2021, Kwok and Atkinson, 1995) estimates for the OH reaction with both investigated ketones, undiscriminating, the contribution of H atom abstraction and OH addition to the olefinic bond to be $k_{abs.} = 0.4 \times 10^{-12}$ cm$^3$ molecule$^{-1}$ s$^1$ and $k_{add.} = 7.8 \times 10^{-11}$ cm$^3$ molecule$^{-1}$ s$^1$, respectively. This suggests that the OH reaction proceeds almost exclusively through the addition to the C=C double bond.

This theoretical result is supported by the findings of the present product studies (see Sect. 3.3). The rate coefficients of 4M3P2 + OH predicted by AOPWIN and determined here experimentally agree within 4 %. However, AOPWIN does not differentiate between certain substitution patterns. Given that there is a good agreement between the results using different references, our results show 4M3P2 to be about 1.25 times more reactive towards OH radicals than 3M3P2. Therefore, the AOPWIN prediction is less accurate in the case of 3M3P2. Estimations for the rate coefficients of the reactions of both ketones with OH

radicals were given as well in an earlier study ($k$(3M3P2 + OH) = $4.2 \times 10^{-11}$ cm$^3$ molecule$^{-1}$ s$^1$ and $k$(4M3P2 + OH) = $5.3 \times 10^{-11}$ cm$^3$ molecule$^{-1}$ s$^1$) based on linear free-energy relationships using ionization potentials (Grosjean and Williams, 1992). Being lower for both ketones, these predictions reproduce though qualitatively the experimentally observed rate coefficient ratio $k$(4M3P2 + OH)/$k$(3M3P2 + OH) $\approx$ 1.26. The same applies when the estimation of the rate constants is performed according to SAR approach by Jenkin et al. (2018).

On the other hand, if the reaction proceeds almost solely via the electrophilic addition to the C=C double bond the reactivity can be examined in terms of the electron density associated with the olefinic bond. We have recently pointed out the importance of defining the appropriate core structure when discussing gas-phase reactivity and related substituent effects in unsaturated oxygenated compounds (Illmann et al., 2021). In the case of the α,β-unsaturated ketones this yields a comparison with structural analogue alkenes where the acetyl moiety is replaced by an H atom, thus *(Z)*-2-butene (for 3M3P2) and

isobutene (for 4M3P2) as structural analogues. Despite the deactivating *(-I)*-effect of carbonyl groups, the ability of the carbonyl group to form hydrogen-bonded intermediates with OH radicals (Smith and Ravishankara, 2002) lead to a presumably higher reactivity of the unsaturated ketones compared to their alkene analogues. Experimental results supporting this assumption were reported previously, although the corresponding alkene were chosen in a different way (Blanco et al., 2012). An attempt to quantify the substituent effects in oxygenated compounds was previously performed by defining a non-

dimensional reactivity factor $x_r = k_{ketone} / k_{alkene}$ (Illmann et al., 2021). By using the latest IUPAC recommendations for the OH radical reactions of isobutene and (Z)-2-butene, respectively, and the results of the present study we obtained for both ketones $x_r > 1$. This is in complete agreement with the expected enhancement of reactivity towards OH radicals. On the other hand, $x_r$ is < 1 for both ketones in the case of the Cl atom reaction. This is not surprising as here the formation of an activating hydrogen-





bonded intermediate is not possible. Therefore, the reactivity towards Cl atoms is determined by the deactivating inductive effect of the carbonyl moiety upon the olefinic bond. However, an in-depth analysis of these effects is more related to fundamentals of physical chemistry and will be included in a separate publication.

**3.2 Cross sections**

Integrated absorption cross sections σ have been determined based on the Beer-Lambert law


$$\int_{v_1}^{v_2} \ln\left(\frac{I_0}{I}\right) dv = \sigma \times c \times l \tag{2}$$

by plotting the integrated absorption band between $v_1$ and $v_2$ against the concentration c. These values are summarised in Table 3 together with literature data where available. Plots used to determine the cross sections are shown in Fig. S1–S2. To the best
of our knowledge, there are no previous reports on the IR cross sections for 3M3P2, 4M3P2 and acetoin in the literature. Profeta and co-workers (2011) reported band intensities for the various absorptions of biacetyl in the gas-phase. The integrated cross section of the carbonyl absorption (1690–1769 cm$^{-1}$) determined within this work is almost identical with their value. The other cross sections agree within < 10 %, where the largest discrepancies are observed for the least intense absorption features (between 870–994 cm$^{-1}$ and 2905–3053 cm$^{-1}$).


**Table 3.** Integrated absorption cross sections determined within this work (resolution: 1 cm$^{-1}$, apodisation: Happ-Genzel, phase correction: Merck, zero-filling: 0) together with available literature references.

| Compound | Range/ cm$^{-1}$ | This work $\sigma \times 10^{18}$ / cm molecule$^{-1}$ | Literature $\sigma \times 10^{18}$ / cm molecule$^{-1}$ | Reference |
|---|---|---|---|---|
| 3-Methyl-3-penten-2-one | 1600–1750 | 35 ± 4 | | |
| 4-Methyl-3-penten-2-one | 1675–1741 | 17 ± 2 | | |
| Acetoin | 3100–2700 | 17 ± 2 | | |
| | 1766–1704 | 14 ± 2 | | |
| Biacetyl | 2905–3053 | 2.7 ± 0.2 | 2.9 | Profeta et al., (2011) |
| | 1690–1769 | 31 ± 2 | 31 | Profeta et al., (2011) |
| | 1392–1476 | 5.1 ± 0.4 | 5.5 | Profeta et al., (2011) |
| | 1297–1392 | 9.3 ± 0.7 | 10 | Profeta et al., (2011) |
| | 1078–1154 | 12 ± 1 | 13 | Profeta et al., (2011) |
| | 870–994 | 3.8 ± 0.4 | 4.2 | Profeta et al., (2011) |
| 2-Hydroxy-2-methylpropanal | 2780–3010 | 30 ± 5 | 10.2 ± 1.6 | Carrasco et al., (2006) |

We recommend to use freshly prepared or purchased samples when working with biacetyl. The investigation of an older sample
stored for several months at temperatures < 8 °C yielded cross sections 30–40 % lower than reported by Profeta et al. (2011)


while the gas-phase IR spectra were identical with the new sample. Thus, it seems likely that degradation takes place even in a cooled sample. However, the absence of foreign absorption features in the FTIR spectra of the older sample remains unexplained.

In order to determine its integrated absorption cross section, 2-hydroxy-2-methylpropanal (HMPr) was generated in situ
through the ozonolysis of 2-methyl-3-buten-2-ol in the presence of sufficient amounts of CO to scavenge any OH radical formed during the reaction. According to the well-established gas-phase ozonolysis mechanism, the initially formed trioxolane will decompose in two possible ways to form either one or the other primary carbonyl, namely formaldehyde (HCHO) and HMPr, and the remaining Criegee intermediate. A secondary formation of both carbonyls resulting from further reactions of the Criegee intermediates is not likely based on the known mechanism. Therefore, the sum of the molar yields of HCHO and
HMPr should yield 100 %. The concentrations of HMPr in each spectrum are thus obtained based on the consumption of the HMPr-precursor 2-methyl-3-buten-2-ol and the molar formation yield of HMPr ($Y_{HMPr}$)

$$Y_{HMPr} = 1 - Y_{HCHO} \qquad\qquad (3)$$

where $Y_{HCHO}$ is the experimentally determined molar formation yield of HCHO. However, in doing so the determined cross section is $(30 \pm 5) \times 10^{-18}$ cm molecule$^{-1}$ which is about 300 % larger than the only available literature reference (Carrasco et al., 2006). If the literature reference is taken as the true value this would either indicate a fundamental underestimation of the HMPr concentration or an erroneous determination of the 2-methyl-3-buten-2-ol concentration in our experiments. The unsaturated alcohol is highly volatile and does not pose quantification problems. More, no wall loss was measured in any of
the chambers either for the alcohol nor for the HMPr (see Sect. 3.3). A falsely determined cross section of the HMPr precursor by a factor up to three is therefore very unlikely. On the other hand, using the literature cross section would result in molar yields of about 190 % for HMPr in our experiments.

Carrasco et al. (2006) synthesised and purified HMPr in the liquid phase and evaporated different amounts of the sample. The concentrations were calculated according to the ideal gas law and subtracting the amounts of formic acid and
formaldehyde still present in the liquid sample despite purification. Although the authors stated that HMPr has also been synthesised in situ, similarly to the present study, the cross section seems to be calculated based solely on the liquid sample evaporation. However, the listed integrated cross section for HCHO, based on the natural logarithm as stated by the authors, differs about a factor of two from other literature references (Nakanaga et al., 1982; Gratien et al., 2007). Besides, the used value is about 2.3 smaller than the cited literature reference (Picquet-Varrault et al., 2002). All this suggest a systematic
conversion error in the previous study. Therefore, we prefer to use the HMPr cross section estimated in our group for the following product study.





### 3.3 Product study of the OH reactions

In the following subsections the terms "α- and β-position" are used with respect to their position related to the carbonyl group.

Thus, the $C_\alpha$ refers to the carbon atom adjacent the carbonyl group. Consequently, the α-$RO_2$ radical names the $RO_2$ where the molecular oxygen added in α-position.

### 3.3.1 3-Methyl-3-penten-2-one

Figure 2 shows evaluation details of IR spectra recorded during a product study experiment of 3M3P2 and used references of

reaction products. The range 3600–2600 $cm^{-1}$ was chosen solely for clarity reasons. More spectra focussing on other spectral ranges can be found in Fig. S3. Figure S4 shows reference spectra for the unsaturated ketones. The remaining absorption features present in the residual spectra after subtraction of 3M3P2, methyl nitrite, methyl nitrate, $HNO_3$, HONO, HCHO, NO and $NO_2$ can be unambiguously attributed to acetoin, biacetyl, acetaldehyde and peroxyacetyl nitrate (PAN), respectively. $CO_2$ formation was clearly observed by examination of its absorption features in the range 2400–2250 $cm^{-1}$ during the irradiation

period. After subtraction of all clearly assigned spectral features the residual spectrum contains only weak absorption bands centred on 1654 $cm^{-1}$, 1364 $cm^{-1}$, 1297 $cm^{-1}$ and 856 $cm^{-1}$. Although a possible indication for the formation of organic nitrates, they are too weak to be reasonably interpreted.

The formation of the identified species can be well explained as first generation products of the OH initiated oxidation of 3M3P2, according to the proposed mechanism depicted in Figure 3. The OH radical is expected to add predominantly to

the C=C double bond in either α- or β-position followed by the addition of $O_2$ to form the corresponding α- or β-$RO_2$ radical (= hydroxyperoxy radical). Their formation and further oxidation pathways (Figure 3) are consecutively defined as $\alpha_n$- and $\beta_n$-pathways, respectively. Due to the presence of NO in excess, under these experimental conditions it is certain that virtually all $RO_2$ radicals will react with NO to mainly form the corresponding RO radicals (= hydroxyalkoxy radicals). The β-RO radical can undergo a bond scission (pathway $\beta_1$, Figure 3) between the α- and β-carbon ($C_\alpha$ and $C_\beta$) to form acetaldehyde and a

hydroxyalkyl radical. This, in turn, reacts with $O_2$ to produce biacetyl. In principle, the H atom abstraction from the β−RO by $O_2$ might lead to the formation of a 2-hydroxy-1,3-dicarbonyl species (pathway $\beta_2$, Figure 3). However, after subtraction of all identified species there are no remaining absorptions to support the occurrence of this pathway. On the other hand, the reaction of $RO_2$ with NO is known to be exothermic, resulting in chemically activated RO radicals potentially prone to "prompt" decomposition (Orlando et al., 2003). This pathway was shown to be important for RO radicals where energy barriers to

decomposition are low, as expected for the β-RO radical according to available estimation methods (Atkinson, 2007; Vereecken and Peeters, 2009). Besides, Atkinson (2007) concluded that decomposition will dominate compared to the reaction with $O_2$ in the case of 1,2-hydroxyalkoxy radicals. This applies also for the thermalized hydroxyalkoxy radicals. All this suggests that pathway $\beta_2$ does either not exist if the precursor β-RO radical is formed through $RO_2$ + NO or its branching fraction is very low.


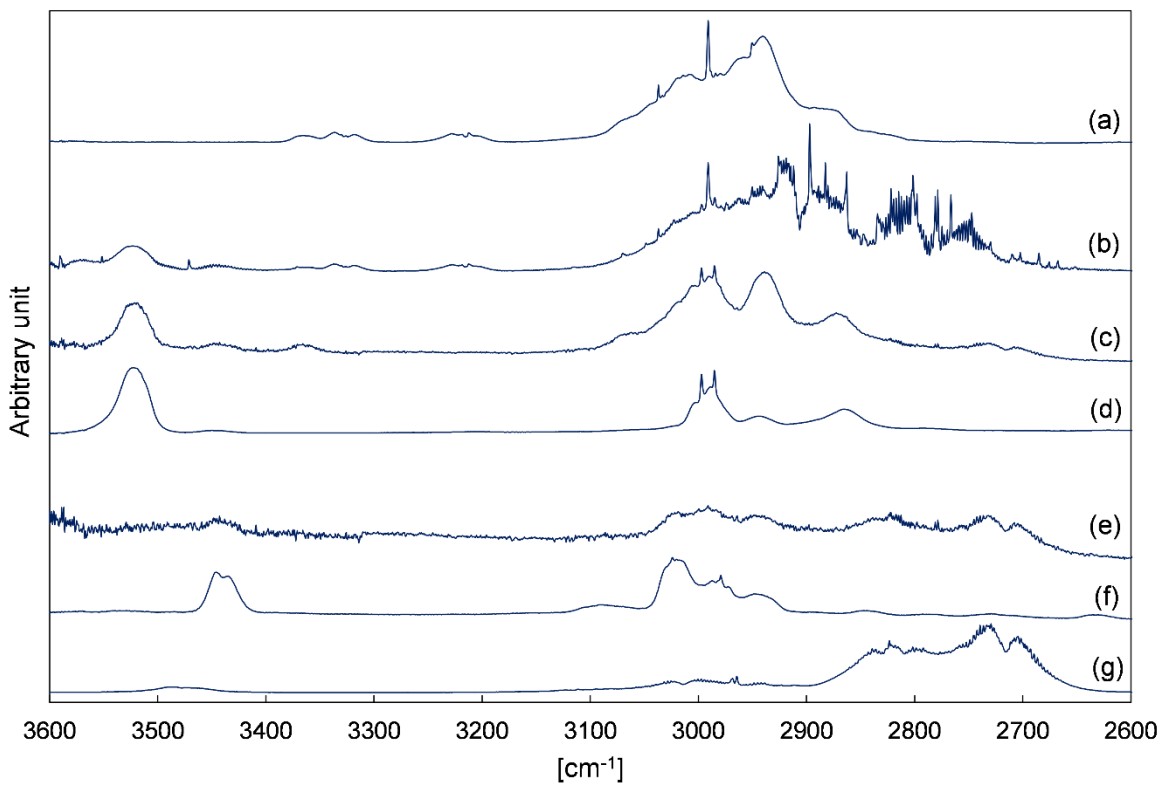

**Figure 2.** Exemplary FTIR spectra of a product study experiment of 3M3P2 + OH: (a) reaction mixture before irradiation, (b) reaction
mixture at the end of the irradiation period, (c) residual spectrum after subtraction of methyl nitrite, methyl nitrate, $HNO_3$, HONO, NO, $NO_2$
and HCHO from (b), (d) reference spectrum of 3-hydroxy-2-butanone (acetoin), (e) residual spectrum after subtraction of 3M3P2 and acetoin
from (c), (f) reference spectrum of 2,3-butanedione (biacetyl), and (g) reference spectrum of acetaldehyde. The spectra are shifted and scaled
individually for a better overview.

410         The addition of the OH radical in β-position followed by the addition of $O_2$ and reaction with NO yields eventually

acetoin (pathways α and $\alpha_1$, Figure 3). The bond scission in the α-RO radical between $C_\alpha$ and the carbon atom of the carbonyl

moiety (pathway $\alpha_1$, Figure 3), yields further acetyl radicals which, after addition of $O_2$, can either form peroxyacetyl nitrate

(through addition of $NO_2$) or react with NO to finally form $CO_2$ and HCHO, under the experimental conditions. The yield of

HCHO related to 3M3P2 oxidation cannot be experimentally determined because it is also produced in the methyl nitrite

photolysis itself. On the other hand, the α-RO radical can also form biacetyl if the bond scission occurs between $C_\alpha$ and $C_\beta$

(pathway $\alpha_2$, Figure 3). In this case acetaldehyde evolves from the simultaneously formed hydroxyalkyl radical due to H atom

abstraction through molecular oxygen.






**Figure 3.** Proposed mechanism for the OH radical initiated oxidation of 3-methyl-3-penten-2-one and further oxidation of the first generation products relevant under the experimental conditions.

**3.3.2 4-Methyl-3-penten-2-one**

Figure 4 depicts evaluation details of IR spectra recorded during a product study experiment of 4M3P2 and references. The spectral range 3200–2600 cm$^{-1}$ was chosen as mentioned above. Acetone, methyl glyoxal and peroxyacetyl nitrate are clearly


identifiable in all residual spectra after subtraction of 4M3P2, methyl nitrite, methyl nitrate, $HNO_3$, HONO, HCHO, NO and $NO_2$. As in the 3M3P2 product studies, the $CO_2$ formation during the irradiation period was unambiguously observed in the

range 2400–2250 cm$^{-1}$ (Fig. S5). Trace (f) in Figure 4 shows an exemplary residual spectrum after subtraction of methyl glyoxal, acetone and PAN references from the spectrum in trace (c). The spectral features are in excellent agreement, in the presented range, with a reference IR spectrum of 2-hydroxy-2-methylpropanal (see Sect. 3.2). This proves its formation in the OH radical initiated oxidation of 4M3P2. After subtraction of all assigned absorptions the remaining residual spectra, in both 480 L and 1080 L product study experiments, contain weak but well-defined absorption bands centred around 3478 cm$^{-1}$,

1722 cm$^{-1}$, 1654 cm$^{-1}$, 1297 cm$^{-1}$ and 856 cm$^{-1}$ (Figure 5). The characteristic absorption pattern of the latter three bands is a strong indication for organic nitrate formation in this reaction system.

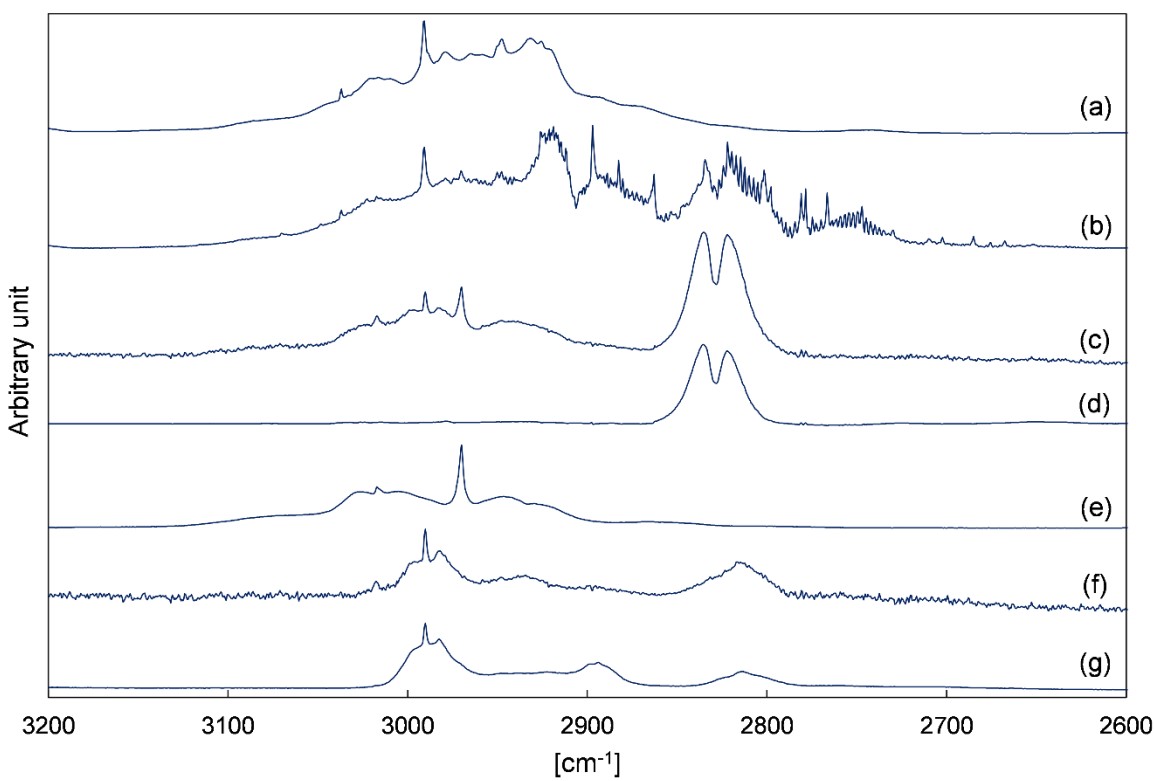

**Figure 4.** Exemplary FTIR spectra of a product study experiment of 4M3P2 + OH: (a) reaction mixture before irradiation, (b) reaction
mixture at the end of the irradiation period, (c) residual spectrum after subtraction of 4M3P2, methyl nitrite, methyl nitrate, NO, $NO_2$ and HCHO from (b), (d) reference spectrum of methyl glyoxal, (e) reference spectrum of acetone, (f) residual spectrum after subtracting methyl glyoxal, acetone and PAN from (c), and (g) reference spectrum of 2-hydroxy-2-methylpropanal generated in situ. The spectra are shifted and scaled individually for a better overview.

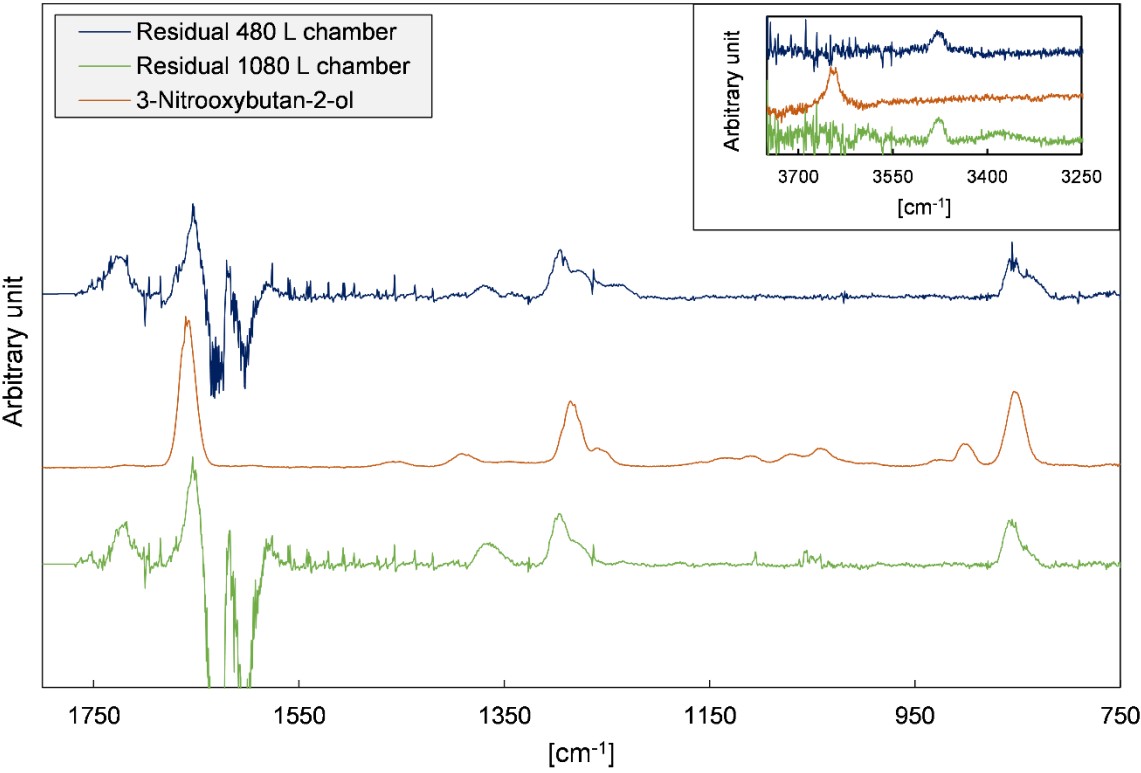

**Figure 5.** Residual spectra of 4M3P2 + OH product study experiments performed in both chambers after subtraction of all identified species and an exemplary reference spectrum of an organic nitrate recorded in our laboratory (Spittler, 2001).

Based on the experimental results obtained here a mechanism for the reaction of OH radicals with 4M3P2 was drawn (Figure 6). The OH radical will add predominantly to either the $\alpha$- or $\beta$-carbon. The subsequent $O_2$ addition will yield the corresponding $\beta$- or $\alpha$-$RO_2$ radical, respectively. As for 3M3P2, all 4M3P2 product studies were conducted under conditions where $RO_2$ radicals are expected to react solely with NO. Acetone can be formed from the conversion of the $\beta$-$RO_2$ with NO into the corresponding $\beta$-RO followed by the scission of the C-C bond between $C_\alpha$ and $C_\beta$ (pathway $\beta_1$, Figure 6). The synchronously generated hydroxyalkyl radical might react with $O_2$ to yield methyl glyoxal. HMPr formation can be explained by a bond scission between $C_\alpha$ and the carbon atom of the carbonyl group (pathway $\alpha_1$, Figure 6). In the present experimental system, the co-generated acetyl radicals yield either PAN or $CO_2$ and HCHO, as discussed above. Methyl glyoxal and acetone can also be formed according to pathway $\alpha_2$ (Figure 6) if the bond scission in the $\alpha$-RO occurs between $C_\alpha$ and $C_\beta$.

Hypothetically, the $\alpha$-RO radical could also produce a 3-hydroxy-1,2-dicarbonyl species (pathway $\alpha_3$, Figure 6) if H atom abstraction via molecular oxygen would occur rather than a C-C bond scission. However, the structure of the weak carbonyl absorption in the residual spectra seems more likely to belong to a single C=O bond rather than a vicinal diketone.




There is thus no indication for the existence of this reaction pathway. Besides, the reaction RO + O₂ is not expected to be competitive to the decomposition channels of the multifunctional RO radical, as discussed above.

**Figure 6.** Proposed mechanism for the OH radical initiated oxidation of 4-methyl-3-penten-2-one and further oxidation of the first generation products relevant under the experimental conditions.

Organic nitrate formation, which is indicated by the residual spectra, is expected to proceed through the reaction of the α-RO₂ or β-RO₂ radical with NO (pathway α_ON or β_ON, Figure 6) followed by the isomerisation of the nascent ROONO adduct (Calvert et al., 2015). RONO₂ formation has also been observed through RO + NO₂ reactions of simple alkoxy radicals (Frost and Smith, 1990; Mund et al., 1998). However, high pressure rate coefficients for these types of reactions are about (1–3) × 10⁻¹¹ cm³ molecule⁻¹ s⁻¹ (IUPAC, current recommendation) whereas for the reactions of RO with O₂ k×[O₂] is about 4 × 10⁴ s⁻¹





according to a recommendation provided by Atkinson (2007). Given that RO + O$_2$ seems even not to compete with unimolecular decomposition for β-hydroxyalkoxy radicals we do not expect RO + NO$_2$ to be experimentally and atmospherically relevant. The absorption features observed in the residual spectra, additionally to the characteristic nitrate

absorptions, indicate the presence of a carbonyl (1722 cm$^{-1}$) and an OH-group (3478 cm$^{-1}$) which can be assigned to multifunctional hydroxycarbonyl nitrates as presented in Figure 6. In the case of the RONO$_2$ species resulting from the α-RO radical, one would expect that intramolecular hydrogen bonding between the OH- and the carbonyl group stabilise the structure. This would cause, on the one hand, a broaden and weaker OH absorption band and, on the other hand, a shift of the carbonyl absorption towards lower wavenumbers. Based on that, it is more likely to tentatively attribute the residual absorptions to the

hydroxycarbonyl nitrate resulting from the β-RO$_2$ radical. The formation of organic nitrates will be further discussed in Sect. 3.3.5.

### 3.3.3 Product yields correction and further oxidation processes

Product yields were obtained by plotting their mixing ratio versus the mixing ratio of consumed unsaturated ketone. The data are corrected only for the wall loss of the unsaturated ketones. These plots are shown in Fig. S6–S7 and exhibit a high linearity

for all identified products except for PAN, CO$_2$ or the sum of PAN and CO$_2$. The latter was used to determine the molar formation yield of acetyl radicals. The non-linearity is a strong indication for further oxidation and secondary processes in the investigated reaction systems leading to acetyl radicals and their further oxidation products. However, this can be well explained by the oxidation of the initially formed reaction products in the OH radical initiated oxidation of the unsaturated ketones. On the other hand, the linearity, observed for the other oxidation products, does not necessarily indicate the absence

of secondary processes. Either secondary formation compensates for loss processes or the scattering of the combined data is larger than the precise non-linearity.

According to Aschmann et al. (2000) the OH initiated oxidation of acetoin proceeds predominantly through alkyl H atom abstraction at the –CH(OH)- entity to yield eventually biacetyl, with a formation yield of about 80 %. This is also expected to be the main loss process under atmospheric conditions. Therefore, the further oxidation of acetoin is an additional source of

biacetyl in the 3M3P2 experimental system. Biacetyl itself is mainly subject to photolysis (R15), under both experimental and atmospheric conditions yielding acetyl radicals.

CH$_3$C(O)C(O)CH$_3$ + hv → 2 CH$_3$C(O)                                                                      (R15)

Acetaldehyde also contributes to the formation of acetyl radicals since the aldehydic H atom abstraction (R16) was shown to account for about 95 % of the OH reaction (Calvert et al., 2011).

CH$_3$C(O)H + OH → CH$_3$C(O) + H$_2$O                                                                      (R16)





Acetone, formed in the oxidation of 4M3P2, will be mainly oxidised by OH radicals through H atom abstraction yielding acetonoxy radicals which readily decompose to HCHO and acetyl radicals (Orlando et al., 2000).

$$CH_3C(O)CH_3 + OH + O_2 + NO \rightarrow CH_3C(O)CH_2O + H_2O + NO_2 \tag{R17}$$
$$CH_3C(O)CH_2O \rightarrow CH_3C(O) + HCHO \tag{R18}$$


However, while being a source of HCHO and acetyl radicals under atmospheric conditions, this reaction cannot play any role in the present experimental set-up, considering the lifetime of acetone with respect to OH. By contrast, HMPr was shown to display a much higher reactivity towards OH, the reaction producing acetone with a yield of unity (Carrasco et al., 2006). Thus, this is a secondary source of acetone in our experiments. Finally, the OH reaction of methyl glyoxal will exclusively

proceed via the abstraction of the aldehydic H atom. The initially formed $CH_3C(O)CO$ radical will readily dissociate into carbon monoxide and $CH_3C(O)$ radicals as well (Green et al., 1990). While photolysis of methyl glyoxal is the main loss process under most atmospheric daytime conditions the OH reaction dominates in the present experimental system.

All these processes interfere in the determination of the intrinsic yield of $CH_3C(O)$ radicals in the reaction of the unsaturated ketones with OH. As discussed above, the further chemistry of acetyl radicals in our experiments may evolve

either into PAN or $CO_2$/HCHO formation. However, in the atmosphere the readily formed acetyl peroxy radical may also react with $HO_2$ radicals to form peroxyacetic acid ($CH_3C(O)OOH$), acetic acid ($CH_3C(O)OH$), $O_3$ and OH radicals (Winiberg et al., 2016). Therefore, in order to evaluate the atmospheric importance of the studied ketones an estimation of the acetyl radicals yield is needed.

The product yields corrected for the secondary reactions mentioned above and wall losses in the simulation chambers

were obtained using the model outlined in Sect. 2.7 and the kinetic parameters listed in Tab. S1 and Tab. S2. The molar yields for both OH reactions are summarised in Table 4 and Table 5. The sum of PAN and $CO_2$ is equal to the molar yield of $CH_3C(O)$ radicals. The errors represent the $2\sigma$ statistical error resulting from the average of all experiments and an additional 10 % relative error to cover further uncertainties derived from the evaluation procedure. Exemplary experimental and simulated time profiles of the ketones and the products are shown in Figure 7 and Figure 8 for both investigated reaction systems.

Acetoin and acetaldehyde are only affected by secondary consumption. Hence, the model estimates an increase for both yields compared to the experimental values of the 3M3P2 oxidation. By contrast, the biacetyl yields exhibit no difference indicating that the loss by photolysis is nearly compensated by the formation through acetoin oxidation. Both acetaldehyde and biacetyl undergo further oxidation forming eventually PAN and $CO_2$. In compliance, the experimental concentration-time-profile for acetyl radicals is only reproduced when introducing significantly lower PAN + $CO_2$ formation yields in the model.

In the 4M3P2 system the modelled acetone yield decreases slightly due to the further oxidation of HMPr. Since methyl glyoxal and HMPr are influenced only by consumption processes, their corrected yields are higher than the experimentally determined. PAN + $CO_2$ correction follows the same pattern as in the case of 3M3P2 + OH.





**Table 4.** Uncorrected and corrected molar yields of the 3M3P2 + OH system.

|  | Acetoin | PAN + $CO_2$ | Biacetyl | Acetaldehyde |
|---|---|---|---|---|
| uncorrected | $0.53 \pm 0.13$ | $0.69 \pm 0.11$ | $0.42 \pm 0.13$ | $0.36 \pm 0.09$ |
| corrected | $0.60 \pm 0.18$ | $0.60 \pm 0.07$ | $0.41 \pm 0.12$ | $0.42 \pm 0.15$ |

**Table 5.** Uncorrected and corrected molar yields of the 4M3P2 + OH system.

|  | Acetone | Methyl glyoxal | HMPr | PAN + $CO_2$ |
|---|---|---|---|---|
| uncorrected | $0.64 \pm 0.11$ | $0.58 \pm 0.13$ | $0.17 \pm 0.05$ | $0.27 \pm 0.04$ |
| corrected | $0.62 \pm 0.09$ | $0.64 \pm 0.16$ | $0.20 \pm 0.05$ | $0.24 \pm 0.06$ |

In all cases the averaged molar yields are in excellent agreement for products expected to be formed in the same reaction channel. Thus, the molar yields of acetaldehyde/biacetyl (pathway $\beta_1$ and $\alpha_2$, Figure 3), acetoin/(PAN + $CO_2$) (pathway $\alpha_1$, Figure 3), and acetone/methyl glyoxal (pathway $\beta_1$ and $\alpha_2$, Figure 6) are nearly the same. In the case of 4M3P2 + OH, the yields of HMPr/(PAN + $CO_2$) formed according to pathway $\alpha_1$ (Figure 6) are still in good agreement within the uncertainties. This supports the absorption cross section of HMPr determined in our study. On the other hand, the results prove that the formation of acetyl radicals can be well quantified by determining the sum of PAN and $CO_2$ in the experimental set-up.

Among $\alpha,\beta$-unsaturated ketones of atmospheric importance only the OH radical initiated oxidation of methyl vinyl ketone (MVK) has been investigated in-depth (Tuazon and Atkinson, 1989; Praske et al., 2015; Fuchs et al., 2018). Tuazon and Atkinson (1989) quantified methyl glyoxal and glycolaldehyde as main oxidation products and concluded that addition of the OH radical to the internal and terminal carbon atom accounts for $28 \pm 9$ % and $72 \pm 21$ %, respectively. This calculation is based on the assumption that the corresponding $\alpha$-RO radical will favour a bond scission between the carbonyl carbon atom and $C_\alpha$ (according to pathway $\alpha_1$ in Figure 3 and Figure 6) due to the much lower predicted energy barrier to decomposition following this pathway (Tuazon and Atkinson, 1989). This has been confirmed by calculations performed by Praske et al. (2015) and is consistent with the SAR provided by Vereecken and Peeters (2009). Assuming $\alpha_1 \gg \alpha_2$ (Fig. 3, 6) the addition of OH according to the $\alpha$- and $\beta$-pathways accounts consequently for $60 \pm 18$ % and $40 \pm 12$ % for 3M3P2 and $26 \pm 8$ % and $74 \pm 22$ % in the case of 4M3P2, respectively, when referenced to the corresponding overall yield. However, at least for 4M3P2 the branching fraction $\alpha_2$ may be important since the estimated energy barrier is lower than for $\alpha_1$ according to the SAR of Vereecken and Peeters (2009). Hence, one should note that the fraction given for the addition to $C_\beta$ ($\alpha$-pathways) represents a lower limit.

Even when the carbon balance is below 100 % for the 4M3P2 + OH reaction addition to $C_\beta$ seems to be much less favoured for 4M3P2 compared to 3M3P2 and MVK.








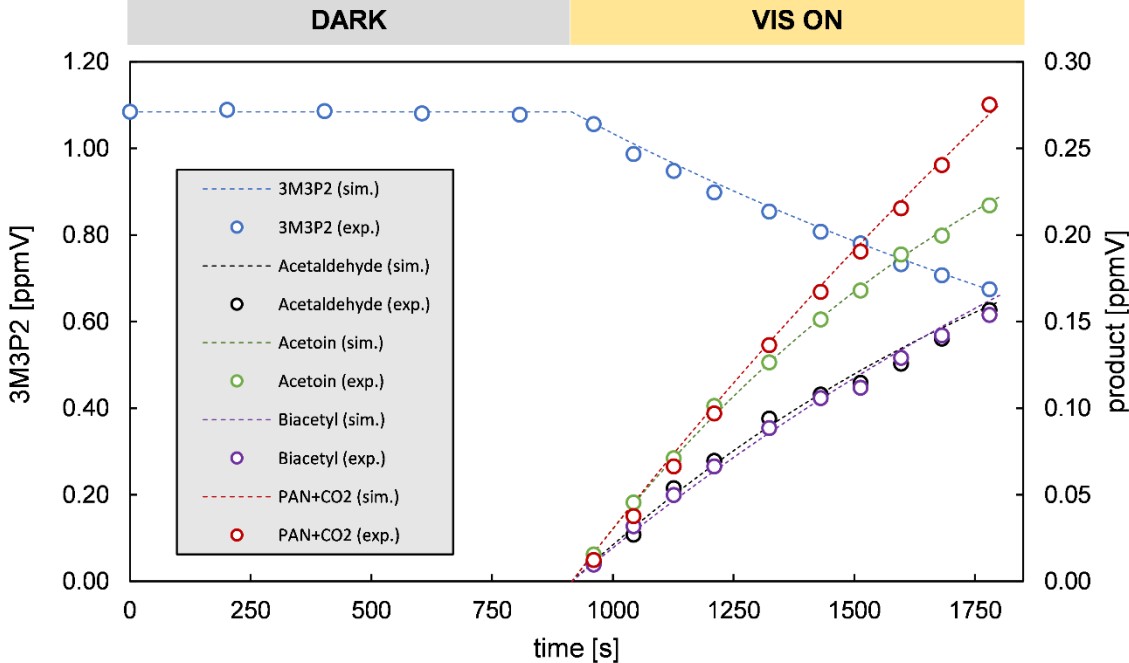

**Figure 7.** Experimental and simulated time profiles obtained for a 3M3P2 + OH experiment.

The stability of alkyl radicals is expected to increase with the degree of substitution ($= d$) due to hyperconjugation. This leads
to the following order for the stability of the alkyl radicals formed by addition of the OH radical to the $\alpha$-position:

$$4\text{M3P2 } (d = 3) > 3\text{M3P2 } (d = 2) > \text{MVK } (d = 1) \tag{4}$$

This is in agreement with the experimentally observed branching ratios. Therefore, while hydrogen bonding should yield a
preference of the addition to the $\beta$-position for all $\alpha,\beta$-unsaturated ketones, as discussed previously with respect to the
reactivity, the observed trend in the branching ratios is possibly related to the stability of the initially formed alkyl radicals.
However, in the case of 4M3P2 the addition to the $\beta$-position could also be sterically hindered.



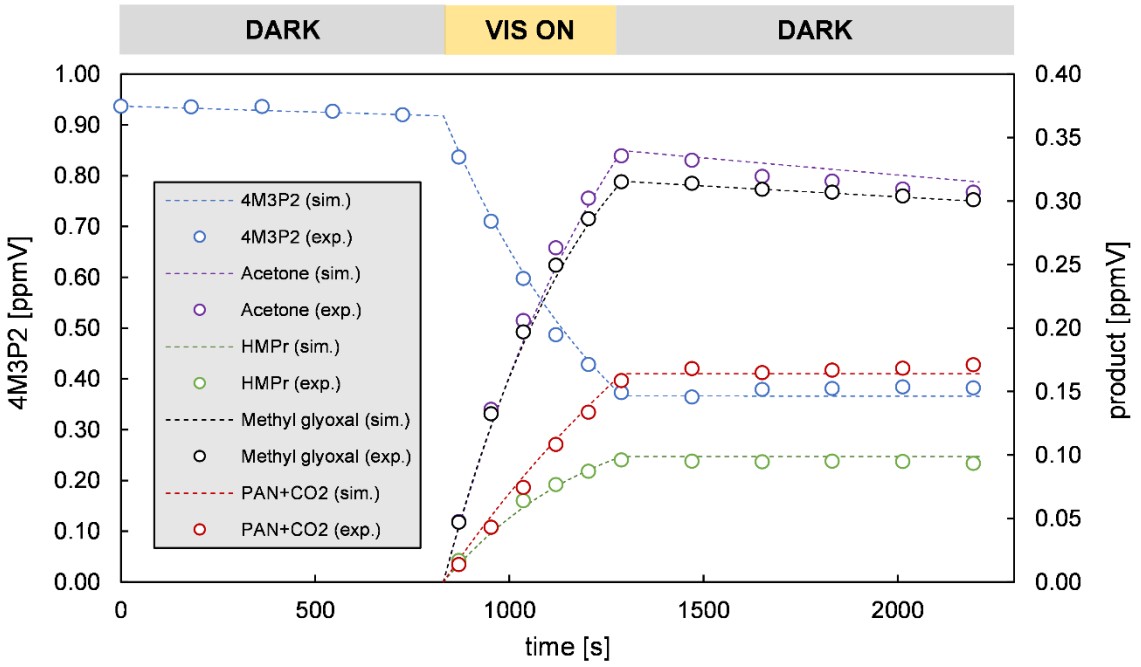

**Figure 8.** Experimental and simulated time profiles obtained for a 4M3P2 + OH experiment.

### 3.3.4 ROONO$_2$ formation

To further elucidate the mechanism, experiments were conducted over a wider range of NO$_2$/NO ratios by adding different amounts of NO$_2$ to the reaction mixture. In all experiments the amount of added NO was sufficient to suppress any ozone formation.

Given that CH$_3$C(O) radicals instantaneously react with oxygen, under the employed experimental conditions, to form the corresponding RO$_2$ radical, their fate should be described by (a) the reaction with NO, (b) the reaction with NO$_2$ to yield peroxyacetyl nitrate (PAN), and (c) the thermal dissociation of PAN to re-generate the RO$_2$ radical.

$$CH_3C(O) + O_2 + M \rightarrow CH_3C(O)OO + M \quad\quad\quad (R19)$$
$$CH_3C(O)OO + NO \rightarrow CH_3C(O)O + NO_2 \quad\quad\quad (R20)$$
$$CH_3C(O)O \rightarrow CH_3 + CO_2 \quad\quad\quad (R21)$$
$$CH_3C(O)OO + NO_2 + M \rightarrow CH_3C(O)OONO_2 + M \quad\quad\quad (R22)$$
$$CH_3C(O)OONO_2 + M \rightarrow CH_3C(O)OO + NO_2 + M \quad\quad\quad (R23)$$



Based on the reactions R20–R23 the ratio PAN/$CO_2$ should only depend on the $NO_2$/NO ratio during the reaction under constant pressure and temperature conditions. Therefore, PAN/$CO_2$ ratios were simulated for various $NO_2$/NO ratios in the experimental temperature range 295–301 K using the IUPAC recommendations for the reactions R20, R22 and R23 (Figure 9). Experimental PAN/$CO_2$ formation ratios were derived from plotting the generated PAN against the formed $CO_2$ during the

irradiation period. The corresponding average $NO_2$/NO ratios were determined by averaging the measured mixing ratios of NO and $NO_2$, respectively, over the same time interval. As can be seen in Figure 9, these data are qualitatively in quite good agreement with the expected ratios based on the model calculations. Hence, the fate of the $CH_3C(O)$ radicals is well-described by the reaction sequence R19–R23 in our experiments.

However, in typical methyl nitrite photolysis experiments the $NO_2$/NO ratios are < 1 if NO is added to supress ozone

formation. Thus, PAN accounts for less than one third of the $CH_3C(O)$ radical's fate (Figure 9). If the experimental set-up does not allow to quantify HCHO or $CO_2$, this leads to a fundamental underestimation of the acetyl radical reaction channels. In this case, addition of $NO_2$ could be useful to favour PAN formation and the determination of the $NO_2$/NO ratio in the experiment could yield an estimation of the PAN/$CO_2$ ratio.

On the other hand, Figure 9 clearly shows the invariance of the PAN + $CO_2$ yield for 4M3P2 under all experimental

conditions, within the uncertainties. There is thus likely no $ROONO_2$ formation from the initially formed α- and β-$RO_2$ radicals since they are either not formed or their thermal dissociation is too large to play any role. This meets one's expectations as lifetimes of alkylperoxy nitrates are in the order of seconds at 298 K and 1 atm and become only relevant at lower temperatures of the upper troposphere (Calvert et al., 2015). However, assuming an average OH concentration of $1 \times 10^6$ $cm^{-3}$ (Bloss et al., 2005) the atmospheric lifetime (with respect to OH) is about 3.4 h thus too short for 4M3P2 to enter higher altitudes. Therefore,

except for PAN, $ROONO_2$ formation does not play any role for 4M3P2.

In the case of 3M3P2 significantly higher yields for PAN + $CO_2$ were observed for higher $NO_2$/NO ratios while being quite consistent at $NO_2$/NO < 1 (Figure 9). This is actually contradicting as additional $ROONO_2$ formation from the initial α- and β-$RO_2$ radicals should lower the overall yields of the main products. However, the molar yields of biacetyl and acetaldehyde are essentially the same as in the other experiments. Thus, the carbon balance exceeds 100 % in the higher

$NO_2$/NO ratio experiments. Unfortunately, these two experiments were the last in a series and it is quite probable that due to wall loading the wall acted as a source of $CH_3C(O)$ radicals during the irradiation. The yield for PAN + $CO_2$ was therefore only given as an average of the experiments with $NO_2$/NO < 1.


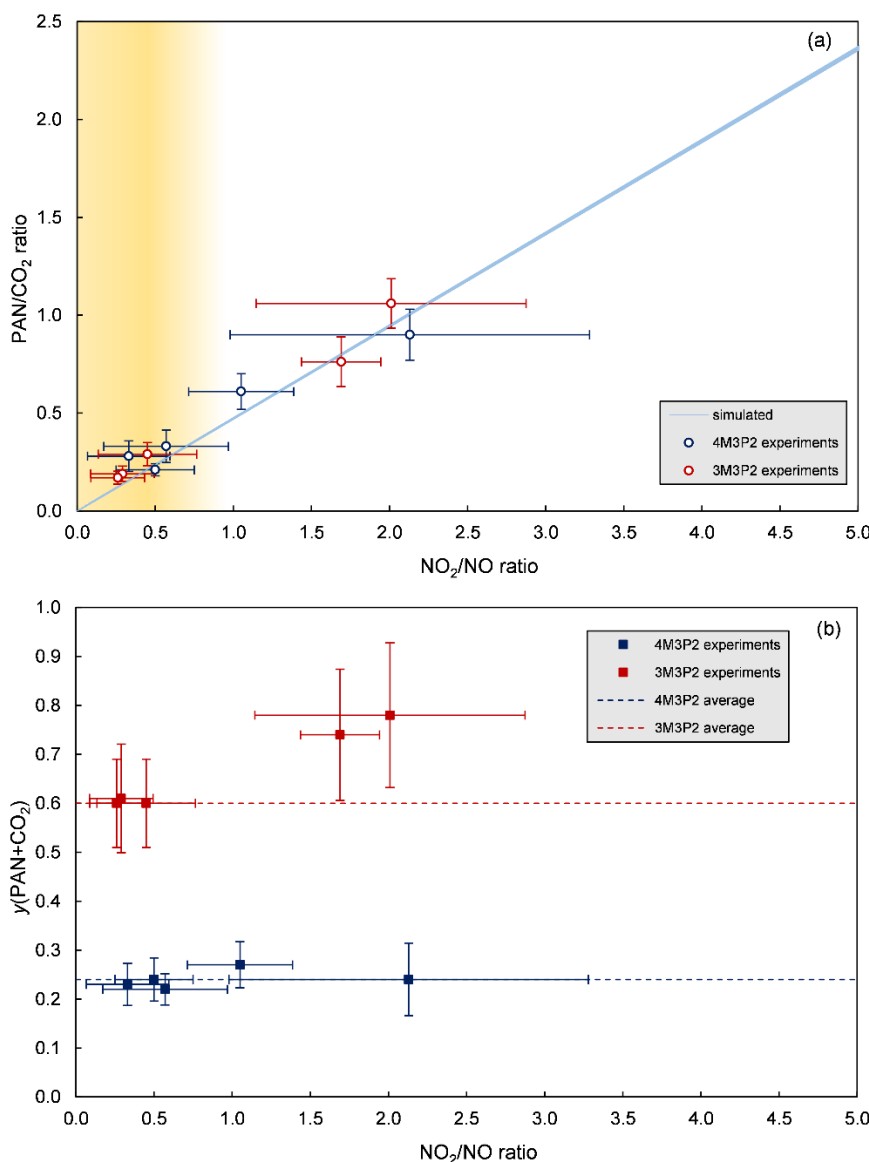

**Figure 9.** (a) Experimentally determined and simulated PAN/CO$_2$ formation ratio as a function of the NO$_2$/NO ratio. (b) Determined molar yields for acetyl radicals (as the sum of PAN and CO$_2$) as a function of the NO$_2$/NO ratio.


### 3.3.5 RONO$_2$ formation

Given that the carbon balance is about one for 3M3P2 and about 0.85 for 4M3P2, respectively, this results in an upper limit of the overall RONO$_2$ yield of 0.15 for 4M3P2 while almost no RONO$_2$ formation occurs in the case of 3M3P2. This is qualitatively in good agreement with the residual spectra clearly indicating the presence of organic nitrates only in the case of



4M3P2. Noda et al. (2000) determined an average absorption cross section of $(1.25 \pm 0.20) \times 10^{-17}$ cm molecule$^{-1}$ (base 10) for the nitrooxy group absorption in the range 800–900 cm$^{-1}$ by averaging available absorption cross sections of organic nitrates. Following this approach, an estimated RONO$_2$ yield of $0.06 \pm 0.03$ is determined which should be regarded as lower limit. While the determined wall loss is in the order of $2 \times 10^{-4}$ s$^{-1}$ the potentially formed RONO$_2$ species could also be subject of significant photolysis and oxidation by OH radicals. A RONO$_2$ yield of about $0.11 \pm 0.03$ has been reported for MVK based

on model-assisted isoprene photooxidation experiments (Paulot et al., 2009). On the other hand, an overall RONO$_2$ yield of about $0.040 \pm 0.006$ has been determined in MVK oxidation experiments (Praske et al., 2015). Thus, the limits (0.06–0.15) reported here for the overall organic nitrate yield are consistent with the data dispersion found in the literature.

       If the branching fractions for the pathways $\alpha_{ON}$ and $\beta_{ON}$ in the 4M3P2 oxidation (Figure 6) were the same this would correspond to a much larger formation yield of the multifunctional $\beta$-RONO$_2$ than $\alpha$-RONO$_2$ simply due to the predominant

addition of the OH radical to C$_\alpha$. This would be consistent with the previous assignment of the residual's spectral features to the $\beta$-RONO$_2$. Former studies on alkenes have shown a structure-dependent RONO$_2$ yield derived from $\beta$-hydroxyperoxy radicals. Accordingly, the nitrate yields resulting from tertiary RO$_2$ were larger than secondary RO$_2$ and as well larger from secondary RO$_2$ comparing to primary RO$_2$, respectively (Matsunaga and Ziemann, 2010). This would indicate $\beta_{ON} > \alpha_{ON}$ in the 4M3P2 oxidation (Figure 6). Besides, Praske et al. (2015) found a two times larger branching ratio for the $\beta$-RONO$_2$ species

resulting from MVK + OH than for the $\alpha$-RONO$_2$. They interpreted this finding in terms of a larger destabilising effect on the initially formed ROONO complex caused by the closer carbonyl group in the case of the $\alpha$-RONO$_2$. Both results further support the assignment of the residual absorptions to the $\beta$-RONO$_2$ species.

       By contrast, both potentially formed RONO$_2$ species in the 3M3P2 oxidation would suffer from a dense chemical environment around C$_\alpha$. Thus, the almost negligible nitrate formation in the case of 3M3P2 is possibly attributed to the steric

hindrance of the hypothetically resulting RONO$_2$ species.

## 4 Atmospheric implications and conclusions

Within this work we determined the rate coefficients for the OH radical and Cl atom-initiated oxidation of 3M3P2 and 4M3P2. This adds to the kinetic information concerning the reaction of these two compounds with NO$_3$ and O$_3$ reported previously (Sato et al., 2004; Canosa-Mas et al. 2005; Illmann et al. 2021; Li et al., 2021) in an effort to complete the gaps in the knowledge

needed for modelling chemistry in the atmosphere. The yields for the identified products formed in both target reactions were found to be independent of the used chamber, the mixing ratios and the light intensity, thus giving confidence in the experimental results.




**Table 6.** Estimated tropospheric lifetimes for the studied α,β-unsaturated ketones. The lifetimes were calculated using the following concentrations: [a] global mean of $1 \times 10^6$ molecule cm$^{-3}$ (Bloss et al., 2005), [b] 12 h average of $3 \times 10^4$ molecule cm$^{-3}$ (Wingenter et al., 1996), [c] 24 h average of $7 \times 10^{11}$ molecule cm$^{-3}$ (Logan, 1985), and [d] 12 h average of $5 \times 10^8$ molecule cm$^{-3}$ (Atkinson, 1991). Rate coefficients were taken from: [e] this work, [f] Illmann et al. (2021), and [g] Canosa-Mas et al. (2005).

| Compound | $\tau(OH)^a$/h | $\tau(Cl)^b$/h | $\tau(O_3)^c$/h | $\tau(NO_3)^d$/h |
|---|---|---|---|---|
| 3M3P2 | 4.3[e] | 33.1[e] | 5.0[f] | 3.6[g] |
| 4M3P2 | 3.4[e] | 29.9[e] | 47.2[f] | 3.9[g] |


Using the kinetic data together with reasonable average concentrations for the respective oxidants (Logan, 1985; Atkinson, 1991; Wingenter et al., 1996; Bloss et al., 2005) allows to estimate tropospheric lifetimes as presented in Table 6, where $\tau(X)$ is the lifetime with respect to the oxidant X calculated according to $1/(k_x[X])$. Both unsaturated ketones did not show measurable photolysis rates in the present experimental set-up. In the atmosphere, their photodissociation lifetimes are

expected to range in the order of days (Mellouki et al., 2015). Consequently, compared to $\tau(OH)$, the photodissociation of both unsaturated ketones is evidently not an important process in the troposphere. The values in Tab. 6 indicate the OH radical as the dominant sink during daytime whereas the $NO_3$ radical plays a similar role at night. However, for 3M3P2 the $O_3$ reaction appears quite competitive during both day and night (Table 6). These estimated lifetimes indicate an oxidative degradation near the emission sources. Nevertheless, a further part of this work indicates that both ketones potentially impact atmospheric

processes on a larger scale due to their huge potential of forming $NO_x$ reservoir species like PAN. In this respect, we have shown that the reaction of OH with both 3M3P2 and 4M3P2 yield $CH_3C(O)$ radicals by prompt decomposition of primary formed RO radicals. In the present work the sum of PAN and $CO_2$ could be used successfully to determine the formation yield of $CH_3C(O)$ radicals in both reaction systems. Moreover, nearly all other identified oxidation products like methyl glyoxal, acetone, acetaldehyde, acetoin and biacetyl are known to generate $CH_3C(O)$ radicals in their further oxidation processes -

simulations based on field data collected worldwide estimate that acetaldehyde, methyl glyoxal and acetone within the troposphere account for about 81 % of the global source for PAN formation (Fischer et al., 2014). Among the oxidation products of both 3M3P2 and 4M3P2 at least methyl glyoxal is a well-known source of secondary organic aerosol in atmosphere (Fu et al., 2008).

     On the other hand, future work is needed to identify clearly the missing products of the OH radical initiated oxidation

of 4M3P2. The absorptions of the residual spectra could be tentatively assigned to the β-RONO$_2$ species. However, other detection methods are necessary for an unambiguous identification and quantification of this species.






*Data availability.* Data can be provided upon request to the corresponding author.

*Author contribution.* NI, RGG, and IPK conducted the experiments and processed the data. NI developed the model and performed the calculations. NI prepared the manuscript with contributions from all co-authors.

*Competing interests.* The authors declare that they have no competing interests.

*Acknowledgements.* The authors gratefully acknowledge funding from the EU Horizon 2020 research and innovation programme through the EUROCHAMP-2020 Infrastructure Activity (grant agreement no. 730997) and the Deutsche Forschungsgemeinschaft (DFG) through the grant agreement WI 958/18-1. R. G. Gibilisco wish to acknowledges the Alexander von Humboldt Foundation for providing a Georg Forster Research Fellowship.

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
