# Peer review of "Atmospheric oxidation of α,β-unsaturated ketones: kinetics and mechanism of the OH radical reaction"

_Atmospheric Chemistry and Physics, 2021_

## Referee Comment (RC1)

Review of "Atmospheric oxidation of α,β-unsaturated ketones: kinetics and mechanism of the OH radical reaction" by Illmann et al. (acp-2021-449)

**General comments:**

This paper describes the rate constants for the reactions of OH radicals with 3-methyl-3-penten-2-one and 4-methyl-3-penten-2-one and the reaction mechanism of these reactions. In addition, the rate constants for the reactions of Cl atoms with 3-methyl-3-penten-2-one and 4-methyl-3-penten-2-one were also investigated. The rate constants were determined by the relative rate method and the relative rates were obtained using three reference compounds for each reaction. The reactant and the products were monitored quantitatively by FTIR and the formation yields of the products were determined, considering their consumption and secondary formation processes. I think that this study was well-conducted and that the reliable data are presented. In addition, the paper is generally well-written. I recommend this paper to be published in Atmospheric Chemistry and Physics after the authors' consideration of my minor comments detailed below.

**Specific comments:**

(1) Page 11, Lines 300–301: Can the authors also discuss which carbon of the C=C double bond the OH radical attacks on preferably? Such the discussion will be useful for the comparison of the results of the product yields.

(2) Page 13, Lines 344–345: Did the authors confirmed that an epoxide is not formed in the ozonolysis of 2-methyl-3-buten-2-ol or that its yield is negligible?

(3) Page 18, Lines 445–446: Show what type of vibrational mode in 3-nitrooxybutan-2-ol the absorption bands at ~1660 cm$^{-1}$, ~1280 cm$^{-1}$, and ~850 cm$^{-1}$ are. I want to know whether those bands are common in both 3-nitrooxybutan-2-ol and β-RONO$_2$ described in Figure 6.

(4) Page 22, Lines 563–564: The methyl group is electron-donating and the acetyl group is electron-withdrawing. So, I think that the OH radical attacks on C$_\beta$ more preferably for 4M3P2 than 3M3P2. But we must consider steric effects, too. The argument in this sentence by the authors is probably based on the assumption of α$_1$ » α$_2$. I am not sure that the assumption is correct in the case of 4M3P2, as the authors mentioned that the branching fraction α$_2$ may be important for 4M3P2.

(5) Page 23, Line 569: I could not understand the meaning of "the stability of alkyl radicals". Explain it.

**Technical comments:**

(1) Page 2, Line42: Since the authors used "Tg" at Line 38, "$10^5$ t" is better to be expressed as "0.1 Tg" or "100 Gg".

(2) Page 2, Line42: Sifniades 2011 → Sifniades et al., 2011

(3) Page2, Line 42: "Hatch et al., 2017" is missing in References.

(4) Page 5, Line 145: Remove "-" between "80-113 s" and "and 15-20 spectra".

(5) Page 6, Line 173: The first "$k_{loss,ref.}$" should be "$k_{loss,ketone}$".

(6) Page 9, Line 260: Add "IUPAC" in References.

(7) Page 17, Line 440: Add "HONO" and "$HNO_3$", as mentioned in the text (Line 428).

(8) Page 22, Line 557: Fig 3, 6 → Figs. 3 and 6

---

## Author Comment (AC1)

**Author's responses to referee comments on:** Illmann et al., Atmos. Chem. Phys. Discuss., https://doi.org/10.5194/acp-2021-449

We thank both referees for the valuable comments on this work. The original comments are shown in black and our responses are marked in blue. Changes made in the text are marked in red in this document as well as in the revised manuscript.

**A. Comments by Referee 1**

General comments:

This paper describes the rate constants for the reactions of OH radicals with 3-methyl-3-penten-2-one and 4-methyl-3-penten-2-one and the reaction mechanism of these reactions. In addition, the rate constants for the reactions of Cl atoms with 3-methyl-3-penten-2-one and 4-methyl-3-penten-2-one were also investigated. The rate constants were determined by the relative rate method and the relative rates were obtained using three reference compounds for each reaction. The reactant and the products were monitored quantitatively by FTIR and the formation yields of the products were determined, considering their consumption and secondary formation processes. I think that this study was well-conducted and that the reliable data are presented. In addition, the paper is generally well-written. I recommend this paper to be published in Atmospheric Chemistry and Physics after the authors' consideration of my minor comments detailed below.

Response: We thank the referee for the positive evaluation and the suggestions.

Specific comments:

(1) Page 11, Lines 300–301: Can the authors also discuss which carbon of the C=C double bond the OH radical attacks on preferably? Such the discussion will be useful for the comparison of the results of the product yields.

Response: The discussion on the branching ratio of the addition of OH to $C_\alpha$ and $C_\beta$ is given in Sect. 3.3.3. We believe that including these thoughts in Sect. 3.1.1 as well would end up in repetition and a circular discussion. Therefore, we believe that it is beneficial to keep the discussion in the mechanistic part of the study.

(2) Page 13, Lines 344–345: Did the authors confirmed that an epoxide is not formed in the ozonolysis of 2-methyl-3-buten-2-ol or that its yield is negligible?

Response: There is no hint for epoxide formation in the gas phase FTIR spectra from 2-methyl-3-buten-2-ol ozonolysis. Besides, to our knowledge, in previous studies on the ozonolysis of 2-methyl-3-buten-2-ol no epoxide formation has been reported. We included the following sentence in Sect. 3.2:

"A secondary formation of both carbonyls resulting from further reactions of the Criegee intermediates is not likely based on the known mechanism. Moreover, by comparison with FTIR spectra of commercially available epoxides, we do not find any hint for epoxide formation in the gas-phase ozonolysis of 2-methyl-3-buten-2-ol, which is in agreement with previous studies (Carrasco et

al., 2007 and references therein). Therefore, the sum of the molar yields of HCHO and HMPr should yield 100%."

Therefore, the following reference was added to the list:

Carrasco, N, Doussin, J. F., O'Connor, M., Wenger, J. C., Picquet-Varrault, B., Durand-Jolibois, R., and Carlier, P: Simulation Chamber Studies of the Atmospheric Oxidation of 2-Methyl-3-buten-2-ol: Reaction with Hydroxyl Radicals and Ozone Under a Variety of Conditions, J. Atmos. Chem., 56, 33–55, https://doi.org/10.1007/s10874-006-9041-y, 2007.

(3) Page 18, Lines 445–446: Show what type of vibrational mode in 3-nitrooxybutan-2-ol the absorption bands at ~1660 cm−1, ~1280 cm−1, and ~850 cm−1 are. I want to know whether those bands are common in both 3-nitrooxybutan-2-ol and β-RONO$_2$ described in Figure 6.

Response: These three IR absorption bands are typical for the -ONO$_2$ moiety of RONO$_2$ compounds, also found in simple alkyl nitrates like methyl nitrate, and result from N=O (1660 cm$^{-1}$, 1280 cm$^{-1}$) and N-O (850 cm$^{-1}$) vibrational modes.

(4) Page 22, Lines 563–564: The methyl group is electron-donating and the acetyl group is electron-withdrawing. So, I think that the OH radical attacks on C$_\beta$ more preferably for 4M3P2 than 3M3P2. But we must consider steric effects, too. The argument in this sentence by the authors is probably based on the assumption of $\alpha_1 \gg \alpha_2$. I am not sure that the assumption is correct in the case of 4M3P2, as the authors mentioned that the branching fraction $\alpha_2$ may be important for 4M3P2.

Response: Yes, the argument is based on this assumption as discussedbefore the lines 563-564. We do not say that this is universally valid. The assumption allows just to derive a lower limit for the addition to C$_\beta$ (α-pathways), which is much lower for 4M3P2 (26%) than for 3M3P2 (60%). Alternatively one can say this indicates the relationship between the branching ratios of the main channels in the reaction with OH radicals is α > β for MVK, α ≈ β for 3M3P2 and α < β for 4M3P2. This trend, however, represents the limiting case $\alpha_1 \gg \alpha_2$.

(5) Page 23, Line 569: I could not understand the meaning of "the stability of alkyl radicals". Explain it.

Response: An alkyl radical is stabilised due to a shift of electron density from a neighbouring σ(C-H)-bond to the p-orbital of the radical carbon (hyperconjugation). Thus the formation of the higher substituted radical will be favoured. The addition of OH to C$_\alpha$ leads to a primary alkyl radical in the case of MVK, a secondary for 3M3P2 and a tertiary in the case of 4M3P2. This should consequently increase the branching ratio for addition of OH to C$_\alpha$ from MVK to 4M3P2 (and decrease the branching ratio for addition of OH to C$_\beta$). The limiting cases of the branching ratios, reported here, follow this trend.

However, combining the referee's comments (1), (4) and (5), we think that the discussion on the branching ratios and related effects was not precise enough and could have been confusing. We therefore modified this part as follows:

"Assuming $\alpha_1 \gg \alpha_2$ (Figs. 3 and 6) the addition of OH according to the α- and β-pathways accounts consequently for 60 ± 18 % and 40 ± 12 % for 3M3P2 and 26 ± 8 % and 74 ± 22 % in the case of 4M3P2, respectively, when referenced to the corresponding overall yield. However, at least for 4M3P2 the branching fraction $\alpha_2$ may be important since the estimated energy barrier is lower than for $\alpha_1$

according to the SAR of Vereecken and Peeters (2009). Hence, one should note that the fraction given for the addition to $C_\beta$ ($\alpha$-pathways) represents a lower limit and an upper limit for the addition of OH to $C_\alpha$ ($\beta$-pathways), respectively. In the limiting case ($\alpha_1 \gg \alpha_2$) this indicates the relationship between the branching ratios of the main channels to be $\alpha > \beta$ for MVK, $\alpha \approx \beta$ for 3M3P2 and $\alpha < \beta$ for 4M3P2 (see Figs. 3 and 6). Due to hyperconjugation the formation of the higher substituted alkyl radical should be favoured. The addition of OH to $C_\alpha$ leads to a primary alkyl radical in the case of MVK, a secondary for 3M3P2 and a tertiary alkyl radical in the case of 4M3P2. Therefore, while hydrogen bonding should yield a preference of the addition to the $\beta$-position for all $\alpha,\beta$-unsaturated ketones, as discussed previously with respect to the reactivity, the observed trend in the branching ratios (in the limiting case $\alpha_1 \gg \alpha_2$) is possibly related to the stability of the initially formed alkyl radicals. In the case of 4M3P2 the addition to the $\beta$-position could also be sterically hindered.

However, one should emphasize that it is not possible to derive the exact branching ratios for $\alpha$ and $\beta$ without deciphering the branching ratios $\alpha_1$ and $\alpha_2$. And attempt to obtain the corresponding rate coefficients for each decomposition channel according to Vereecken and Peeters (2009) failed since the calculated branching ratios for $\alpha_1$ and $\alpha_2$ are about 0.1 and 0.9, respectively, in the case of 4M3P2 which is contradicted by the observed first-generation yields of 2HMPr and PAN + $CO_2$. Vereecken and Peeters (2009) stated the accuracy of the predicted rate coefficients to be within a factor of 5–10. Therefore, it is not possible to derive any further statement on the ratio $\alpha_1 : \alpha_2$.

Technical comments:

(1) Page 2, Line42: Since the authors used "Tg" at Line 38, "$10^5$ t" is better to be expressed as "0.1 Tg" or "100 Gg".

Response: We replaced "$10^5$ t" with "0.1 Tg", accordingly.

(2) Page 2, Line42: Sifniades 2011 → Sifniades et al., 2011

Response: Corrected accordingly.

(3) Page2, Line 42: "Hatch et al., 2017" is missing in References.

Response: We added the missing reference.

(4) Page 5, Line 145: Remove "-" between "80-113 s" and "and 15-20 spectra".

Response: We removed "-" and put "," instead.

(5) Page 6, Line 173: The first "$k_{loss,ref.}$" should be "$k_{loss,ketone}$".

Response: Corrected accordingly.

(6) Page 9, Line 260: Add "IUPAC" in References.

Response: We added the following references related to the IUPAC recommendations:

Atkinson, R., Baulch, D. L., Cox, R. A., Crowley, J. N., Hampson, R. F., Hynes, R. G., Jenkin, M. E., Rossi, M. J., Troe, J., and IUPAC Subcommittee: Evaluated kinetic and photochemical data for atmospheric chemistry: Volume II – gas phase reactions of organic species, Atmos. Chem. Phys., 6, 3625–4055, https://doi.org/10.5194/acp-6-3625-2006, 2006.

Mellouki, A., Ammann, M., Cox, R. A., Crowley, J. N., Herrmann, H., Jenkin, M. E., McNeill, V. F., Troe, J., and Wallington, T. J.: Evaluated kinetic and photochemical data for atmospheric chemistry: volume VIII – gas-phase reactions of organic species with four, or more, carbon atoms ($\geq C_4$), Atmos. Chem. Phys., 21, 4797–4808, https://doi.org/10.5194/acp-21-4797-2021, 2021.

We noted that, unfortunately, the latest data sheets given in the supplement of Mellouki et al. (2021) are not present on the IUPAC webpage whose sheets we used for the calculations. Given that the updated sheet for isobutene recommends a smaller error than previously, we had to recalculate the error assigned to our rate coefficient when using isobutene listed in Tab. 2.

(7) Page 17, Line 440: Add "HONO" and "HNO3", as mentioned in the text (Line 428).

Response: Corrected accordingly.

(8) Page 22, Line 557: Fig 3, 6 → Figs. 3 and 6

Response: Corrected accordingly.

**B. Comments by Referee 2**

This is a very nice laboratory experimental study of the kinetics and mechanism of the reactions of OH with 3-methyl-3-penten-2-one and 4-methyl-3-penten-2-one. Kinetic data on Cl + the same two unsaturated ketones is also presented. The experiments and analysis are of high-quality (though the non-use of standard numerical integration tools is surprising) and clearly presented. I have only minor comments, which are listed below.

Response: We thank the referee for the positive evaluation and the suggestions.

L21     RONO2 is one type of organic nitrate but so is PAN. Perhaps simply write "Based on the calculated product yields an upper limit of 0.15 was determined for the yield of RONO2......"

Response: We modified the sentence accordingly.

L37     isoprene is not the most abundant NMHC (as it is very reactive) but has the highest emission strength

Response: We modified the sentence as follows:

"…mainly through the gas-phase oxidation of isoprene, the NMHC which is most abundantly emitted into the atmosphere, with an estimated annual emission up to 750 Tg (Calvert et al., 2000; Guenther et al., 2006)."

L45    can you provide an estimate of the impact of the loss of $\alpha,\beta$-unsaturated ketones on ozone and SOA formation ?? my guess is that it is not significant.

Response: The POCP estimates for the studied ketones might be relatively high calculated only upon the reactivity towards OH radicals. A comprehensive estimation, as well as the SOA formation potential implies information that was not acquired in the present study. Therefore, we consider that such estimates are presently highly speculative and would prefer not to include them in the paper.

L47    replace "proving" with "identifying" ?

Response: The replacement became redundant since we modified the paragraph (see next comment).

L48      under which (NOx) conditions are formaldehyde and methyl glyoxal the main oxidation products?

Response: We intended here to summarise the mechanism when the MVK derived peroxy radicals react with NO. To be more precise, we extended and rephrased this paragraph as follows:

"Up to date, only the OH radical reaction of MVK has been intensively studied (Tuazon and Atkinson, 1989; Galloway et al., 2011; Praske et al., 2015; Fuchs et al., 2018). Under high-NO conditions, where virtually all peroxy radicals react with NO, glycolaldehyde and methyl glyoxal together with formaldehyde and PAN were identified as first-generation products. Praske et al. (2015) found also a low $RONO_2$ yield of about 4%."

We included also a missing reference dealing with the MVK oxidation:

Galloway, M. M., Huisman, A. J., Yee, L. D., Chan, A. W. H., Loza, C. L., Seinfeld, J. H., and Keutsch, F. N.: Yields of oxidized volatile organic compounds during the OH radical initiated oxidation of isoprene, methyl vinyl ketone, and methacrolein under high-$NO_x$ conditions, Atmos. Chem. Phys., 11, 10779–10790, https://doi.org/10.5194/acp-11-10779-2011, 2011.

L56-61   This text, describing a method that is not used, should be removed.

Response: With due respect, the text is delivering an explanation for the need to develop a proper method, which allows yields corrections when target species have secondary sources in the experimental system. Therefore, we consider it necessary for understanding reasons.

L67    PAN levels depend on the temperature and levels of e.g. NO, but not PAN formation

Response: We are not sure if we understand the comment correctly. What is meant in the paper by "PAN formation" is not the isolated reaction of $CH_3C(O)O_2 + NO_2$ but the net PAN formation (= the steady-state level) in the system. Without continuous production one would expect that PAN decomposes thermally and the $CH_3C(O)O_2 + NO$ reaction would act as a loss process. This, in turn,

depends on the temperature and the ratio of $NO_2/NO$, as we show in Sect. 3., since it is determined, besides the thermal decomposition of PAN, solely by the ratio of the rates of the reactions of $CH_3C(O)O_2 + NO_2$ and $CH_3C(O)O_2 + NO$.

L74    In the Table, the reference column needs to be altered so that it is clear what the "this work" references actually refer to

Response: We tried to modify the table. However, with the specification imposed by ACP we do not find a better solution.

L90    "Cleanliness is proved by FTIR" ? Perhaps "purity was confirmed by FTIR" is better.

Response: In our opinion "Cleanliness" is correct since it refers to the simulation chamber. For clarity the text is completed:

"Cleanliness of the chamber is proved by FTIR. "

L112    "Reactants and products are basically monitored using in-situ FTIR spectroscopy". Delete "basically".

Response: Deleted.

L148    suppress

Response: Corrected.

L151    replace "infolds" with  "contains"

Response: In our opinion, the term is used here properly.

L173    The assumption that the wall loss rate is the same when the lights are on and when the lights are off should be mentioned. With fluorescent lamps (which get warm when on) this is often not the case as the glass walls are heated during operation which leads to convection and thus more rapid transport of gases to the walls.

Response: We checked the wall losses under both "dark" and "light" experimental conditions and the difference was not measurable. Besides, due to the construction details, the lamps casings are, in both chambers, cooled with air thus preventing an excessive heating of the reactors' walls. Instead of including the assumption in Sect. 2.5 we included our observation in the beginning of Sect. 3:

"Irreversible first-order wall losses of 3M3P2 and 4M3P2 were found to be negligible in the 480 L chamber and in the range $(1-6) \times 10^{-5}\,s^{-1}$ and $(1-8) \times 10^{-5}\,s^{-1}$ in the 1080 L chamber, respectively. In separate control experiments, containing the target species in the bath gas only, no difference was observed for the wall loss in the dark and when the mixture was irradiated over the typical length of

an experiment. Hence, an increased wall loss rate due to convection induced by heated walls could be ruled out."

L179 "An average value of the cross sections given by Profeta et al. (2011) and Talukdar et al. (2011) has been used for methyl glyoxal". Please justify this. How different are the results of the two studies cited ?

Response: The results are almost identical and agree within <3%. We did not want to give a preference to one of the studies. By contrast, other studies report values differing up to about 30%. However, they do not report experimental details. Based on the excellent agreement between the two cited studies and the experimental details given, we consider them trustworthy. We modified this as follows:

"For methyl glyoxal we used the cross sections determined by Talukdar et al. (2011). When employing the values reported by Profeta et al. (2011) the obtained mixing ratios of methyl glyoxal are, however, almost identical."

L191  2.7 Modelling. This is a peculiar (outdated) approach to the problem. It would be interesting to know why none of the commonly used numerical integration programs were used such as KINTECUS (freeware for academia). I encourage the authors to recheck their results using such a program.

Response: In this study modelling was used solely for the purpose of correcting the product yields for secondary sources and sinks and not for describing the reaction systems in extenso. The detail gaps we intend to fill through this study would eventually lead to a comprehensive mechanism that can be checked by available numerical integration programs or even integrated in the larger mechanistic schemes (MCM...). However, this is subject of future work.

L207  "loss" = loss rate ?

Response: We added "rate".

L222  "If pseudo-first order conditions are proven by the experimental data...." Please indicate how this is evaluated (exponential decay ??)

Response: For clarity we modified the sentence as follows:

"If the experimental logarithmic decay of the concentration of A demonstrates pseudo-first order conditions, a constant OH concentration is included based on the consumption of A during the irradiation."

L317  3.2 Infra-Red cross sections

Response: Since the work was performed solely by means of FTIR, we thought that this addition is somewhat redundant. Nevertheless, we changed the heading accordingly.

L354 "more" = "moreover"

L373   3.3.1 3-Methyl-3-penten-2-one + OH

Response: Since this is a subsection of 3.3 (Product study of the OH reactions) this is addition might be redundant. Nevertheless, we changed the heading accordingly.

L374   "Figure 2 shows evaluation details of IR spectra...." Delete "evaluation details"

L374   replace "product study experiment of 3M3P2" with "during an experiment to examine product formation in the OH-initiated oxidation of 3M3P2"

Response: We believe that deleting "evaluation details" could be confusing since Fig. 2 shows evaluated spectra (after subtraction of certain species) and not only spectra recorded during an experiment. For clarity, we rephrased the sentence as follows:

"Figure 2 shows details obtained by evaluating IR spectra recorded during a 3M3P2 + OH experiment and the references used to identify the reaction products."

L392   replace "no remaining absorptions" with "no remaining IR absorption features"

Response: We rephrased the sentence as follows:

"However, after subtraction of all identified species there are no remaining IR absorption bands to support the occurrence of this pathway."

L425   replace "3.3.2 4-Methyl-3-penten-2-one" with "3.3.2 4-Methyl-3-penten-2-one + OH"

Response: Since this is a subsection of 3.3 (Product study of the OH reactions) this is addition might be redundant. Nevertheless, we changed the heading accordingly.

L496   replace "under both experimental and atmospheric conditions" with "in the present experiments and in the atmosphere"

Response: We modified the sentence as follows:

"Biacetyl itself is mainly subject to photolysis (R15), under both the present experimental and atmospheric conditions, yielding acetyl radicals."

L516   "While photolysis of methyl glyoxal is the main loss process under most atmospheric daytime conditions the OH reaction dominates in the present experimental system.". Please do the calculation and compare J-CH3C(O)CHO with k(MGLY)*[OH] for the present experiments.

Response: This statement was based on the comparison. For clarity, we included the following sentence:

"Based on the lamp spectrum we calculated the photolysis frequency of methyl glyoxal for all experimental conditions. The ratio between $k$(CH$_3$C(O)CHO + OH) × [OH] and J(CH$_3$C(O)CHO) was found to be typically > 7. Hence, while photolysis of methyl glyoxal is the main loss process under most atmospheric daytime conditions the OH reaction dominates in the present experimental system."

L612   "become only relevant at" . What does this mean ? At what temperature would a non-negligible fraction of ROONO$_2$ be present ?

Response: In the case of ethylperoxy nitrate, the lifetime with respect to thermal decomposition is around 0.2 s at 298 K and exceeds several days, for instance, at 210 K, allowing processes other than thermal decomposition to dominate, as discussed in Calvert et al. (2015). However, since the lifetime of the unsaturated ketones with respect to OH is quite short, these species cannot reach higher altitudes (and thus regions with temperatures allowing the formation of non-negligible fractions of ROONO$_2$). To avoid misunderstandings, we slightly modified the following paragraph:

"This meets one's expectations as, according to literature references (Calvert et al., 2015), the lifetimes of alkylperoxy nitrates are in the order of seconds at 298 K and 1 atm and so they become relevant as NOx reservoir when formed at lower temperatures, encountered in the upper troposphere. However, assuming an average OH concentration of $1 \times 10^6$ cm$^{-3}$ (Bloss et al., 2005) the atmospheric lifetime of 4M3P2 (with respect to OH) is about 3.4 h, thus too short for it to reach higher altitudes. ..."

L638   "the potentially formed RONO2 species could also be subject of significant photolysis". Please assess this properly. What are the cross-sections of RONO2 at the photolysis wavelengths likely to be. As CH3ONO is used as OH source, there is presumably good overlap with the lamp-spectra. The same applies to the loss via OH. What do you expect the loss rate to be for the available OH concentration ?

Response: Based on textbook knowledge on the UV spectra (The MPI-Mainz UV/VIS Spectral Atlas) of nitrites and nitrates species, the absorption maximum for nitrates is shifted towards lower wavelengths (275-340 nm) than those of nitrites (320-400 nm). Carbonyl nitrates (e.g. nitrooxy butanones) exhibit cross sections in the range of $10^{-19} – 10^{-20}$ cm$^2$/molec. However, since we do not know the precise structure of the RONO$_2$, we can only speculate on the contribution of photolysis in our system. The formulation we used could have led to misunderstandings. We intended to say that we expect rather photolysis than further oxidation by OH. For some carbonyl nitrates (of different structure), it was shown that photolysis dominates over the OH reaction. For clarity and to avoid confusion, we deleted "significant" and included the following paragraph:

"While the determined wall loss is in the order of $2 \times 10^{-4}$ s$^{-1}$ the potentially formed RONO$_2$ species could also be subject of photolysis and oxidation by OH radicals. SAR methods for the prediction of OH rate coefficients were shown to fail at carbonyl nitrates (Suarez-Bertoa et al., 2012). Therefore, Suarez-Bertoa et al. (2012) proposed alternative substituent factors optimized for carbonyl nitrates in which the factor for the –ONO$_2$ group is less deactivating than in other SAR approaches. Applying this factors to the SAR of Kwok and Atkinson (1995) yields predicted rate coefficients of $1.3 \times 10^{-12}$ cm$^3$

molecule$^{-1}$ s$^{-1}$ and 5 × 10$^{-13}$ cm$^3$ molecule$^{-1}$ s$^{-1}$ for the α-RONO$_2$ and β-RONO$_2$, respectively, which would correspond to loss rates of about 1.3 × 10$^{-5}$ s$^{-1}$ and 5 × 10$^{-6}$ s$^{-1}$, respectively, due to the OH reaction. For α- and β-carbonyl nitrates it was shown that photolysis dominates over the OH initiated oxidation (Suarez-Bertoa et al., 2012; Picquet-Varrault et al., 2020). Hence, by comparison with available data (Suarez-Bertoa et al., 2012; Picquet-Varrault et al., 2020) larger loss rates result likely from photolysis of the RONO$_2$ species in our experiments. However, we believe that any further statement would be highly speculative."

Therefore, the following references were added:

Picquet-Varrault, B., Suarez-Bertoa, R., Duncianu, M., Cazaunau, M., Pangui, E., David, M., and Doussin, J.-F.: Photolysis and oxidation by OH radicals of two carbonyl nitrates: 4-nitrooxy-2-butanone and 5-nitrooxy-2-pentanone, Atmos. Chem. Phys., 20, 487–498, https://doi.org/10.5194/acp-20-487-2020, 2020.

Suarez-Bertoa, R., Picquet-Varrault, B., Tamas, W., Pangui, E., and Doussin, J.-F.: Atmospheric Fate of a Series of Carbonyl Nitrates: Photolysis Frequencies and OH-Oxidation Rate Constants, Environ. Sci. Technol., 46, 12502–12509, https://doi.org/10.1021/es302613x, 2012.

L653   "would suffer from a dense chemical environment around Cα". I've no idea what this statement means. Please re-phrase.

Response: We reformulated the sentence as follows:

"By contrast, both potentially formed RONO$_2$ species in the 3M3P2 oxidation would contain a quaternary C$_\alpha$ atom surrounded by bulky substituents. Thus, the almost negligible nitrate formation in the case of 3M3P2 is possibly attributed to the steric hindrance of the hypothetically resulting RONO$_2$ species."